# Food Label Readability and Consumption Frequency: Isolating Content-Specific Effects via a Non-Equivalent Dependent Variable Design

**DOI:** 10.3390/nu18020197

**Published:** 2026-01-07

**Authors:** Constanza Avalos, Nick Shryane, Yan Wang

**Affiliations:** Department of Social Statistics, University of Manchester, Manchester M13 9PL, UK; N.Shryane@manchester.ac.uk (N.S.); yan.wang-5@manchester.ac.uk (Y.W.)

**Keywords:** food labeling, food consumption, nutrition surveys, ultra-processed food, readability, salience

## Abstract

Objective: This study investigates the association between consumers’ perceived readability of Multiple Traffic Light (MTL) label print size—a theoretical structural gatekeeper for visual salience—and self-reported food consumption frequency in the United Kingdom. We aimed to disentangle the effect of label readability from label content. Using non-equivalent dependent variables (NEDVs), we tested whether the association is specific to unhealthy convenience foods and absent for healthy or unlabeled foods, while also examining heterogeneity across consumer subgroups. Methods: Data from 8948 adults across four waves (2012–2018) of the UK Food and You Survey were analyzed. Cumulative link ordinal logistic regressions were employed to model the association between self-reported print size readability and the consumption frequency of four product types: pre-packaged sandwiches and pre-cooked meat (unhealthy, labeled targets), dairy (nutritionally advisable, labeled control), and fresh meat (unlabeled control). Models were adjusted for sociodemographic covariates, health behaviors, and survey wave fixed effects. Results: The findings reveal a content-specific and significant dynamic relationship exclusively for pre-packaged sandwiches. In 2012, a one-unit increase in readability was associated with a 9% decrease in the odds of frequent consumption (OR=0.91), consistent with a warning effect. However, by 2018, this relationship reversed to a 4% increase (OR=1.04), indicating that higher readability became associated with more frequent consumption. In contrast, a persistent null association was observed for pre-cooked meat, dairy, and fresh meat. Subgroup analyses for sandwiches indicated that the association with readability was strongest among less-engaged consumers. Conclusions: Empirical evidence challenges the utility of a standardized approach to food labelling. The results suggest that the effectiveness of label salience is contingent not just on the consumer but on the product’s context and the content of its message, highlighting the need for adaptive rather than uniform policy standards.

## 1. Introduction

Diet-related health issues remain a critical challenge in the United Kingdom. Ultra-processed foods (UPFs) now account for over half of dietary energy intake [1,2]. Among these, Ready-to-Eat Meals (REMs) represent a rapidly growing category. This sector has achieved 90% market penetration [3]. Unlike snacks or ingredients, REMs are consumed as complete meal replacements [4,5]. Consequently, their nutritional profile significantly impacts overall diet quality. However, convenience often masks poor nutritional quality. REMs frequently contain high levels of saturated fat, sodium, and sugar [6,7], increasing cardiovascular disease risk [8]. High consumption frequency and nutritional homogeneity make REMs an ideal case study. They are well-suited for evaluating front-of-pack (FOP) labelling interventions aimed at signaling health risks.

The UK government introduced the Multiple Traffic Light labelling policy in 2013 (Figure 1). Self-reported understanding is high [9]. Yet market data reveal only modest shifts in REM purchasing behavior [10]. A distinction between label presence and label readability offers a plausible explanation. Current UK Food Standards Agency guidelines mandate a minimum font size of just 1.2 mm [11]. A label may be visually salient due to color. Nevertheless, it fails as a communicative tool if the text falls below a consumer’s perceptual threshold.

Clarifying label effectiveness requires distinguishing four theoretical constructs. Salience refers to initial attentional capture, often driven by color or size [12]. Readability describes the perceptual ease of decoding text [13]. Objective understanding measures the accuracy of nutritional interpretation [14]. Finally, subjective understanding is the consumer’s feeling of being informed [15]. Objective understanding is crucial for nutritional literacy. However, consumer choice models suggest that subjective understanding is often the more proximal driver of purchasing decisions [16]. Color generates salience, but readable text is necessary for cognitive decoding. If text falls below the perceptual threshold, the pathway to subjective understanding is blocked. This renders the warning ineffective regardless of visual prominence.

We propose a salience-to-understanding mechanism to synthesize this process (Figure 2). To influence behaviour, a label must traverse a pathway from visual detection to subjective understanding [17]. This feeling of being informed predicts choice. Theoretical frameworks suggest this pathway is not uniform. It is likely moderated by consumers’ pre-existing health beliefs and motivations [18]. The Health Belief Model posits that highly engaged consumers are already sensitized to risk information. They may use labels merely to confirm their choices [19]. In contrast, perceptual clarity may be the critical bottleneck for less-engaged consumers. Without high readability, the warning signal may be ignored. Thus, we expect substantial heterogeneity across consumer subgroups.

Evaluating this mechanism poses a methodological challenge: distinguishing readability from content. A legible label does not intrinsically deter consumption. Rather, it intensifies the conveyed message. For unhealthy REMs, enhanced readability should strengthen the warning signal and theoretically curb consumption. In contrast, readability may reinforce a positive signal for healthier products [20]. This could boost consumption or leave it unaffected. Conventional observational analyses may confound these opposing effects. Moreover, standard quasi-experimental methods like Difference-in-Differences (DiD) prove impractical. The policy’s simultaneous nationwide rollout lacks arbitrary exposure assignment rules [21].

This study implements a Non-Equivalent Dependent Variable (NEDV) design to overcome these challenges [22]. We leverage repeated cross-sectional data from the UK Food and You Survey. We isolate the effects of label readability by contrasting target outcomes (unhealthy REMs) with control outcomes (dairy and fresh meat). These controls share exposure to common market trends. However, they remain theoretically insulated from the warning mechanism. Dairy typically features green/amber codes, while fresh meat carries no labels.

This research investigates the association between perceived readability of MTL print size and food consumption frequency. It specifically tests the hypothesized salience-to-understanding pathway through three hypotheses:H1: Perceived readability will have a significant, dynamic association with the consumption of unhealthy REMs (sandwiches/pre-cooked meat), reflecting a response to the warning signal.H2: Perceived readability will show a null or reinforcing association for the healthy labeled control (dairy) and a null association for the unlabeled control (fresh meat). This confirms the effect is not a result of general consumer vigilance.H3: The association between readability and consumption will be strongest among less-engaged consumer subgroups, for whom label salience acts as a primary informational cue rather than a confirmation of pre-existing beliefs.

## 2. Materials and Methods

### 2.1. Study Design

This study employs a quasi-experimental Non-Equivalent Dependent Variable (NEDV) design (Figure 3) [22,23,24]. A primary challenge for evaluating this policy was that the MTL intervention was implemented across the entire United Kingdom simultaneously in 2013. Because the whole population was exposed at once, it was impossible to establish a conventional control group of unexposed individuals [25].

Consequently, standard quasi-experimental methods that rely on comparing treated and untreated populations, such as Difference-in-Differences (DiD), were not feasible [21]. Similarly, an Interrupted Time Series (ITS) design was rejected due to the low temporal resolution of the available biannual data, which precludes the robust modelling of trends required to detect a specific policy step change [26]. To overcome these limitations, we adopted the NEDV design, which constructs a valid counterfactual using outcome variables (comparable food products) rather than population groups (people) [23].

A NEDV is an outcome variable theoretically insulated from the intervention but responsive to potential confounders, such as local history or selection bias [27,28]. Causal inference is strengthened if the effect manifests in the target dependent variable but is absent in comparable NEDVs, thereby ruling out generic biases [29].

The target outcome is the consumption frequency of REM products (pre-packaged sandwiches and pre-cooked meat), which are theoretically responsive to red warning labels [7,30]. We selected two NEDVs to control for market and behavioral confounders. Dairy products (labeled control) share similar retail environments and industry strategies (e.g., reformulation) but typically display reinforcing green/amber codes rather than warnings [1,31]. Fresh meat (unlabeled control) captures general consumer trends but remains isolated from the specific MTL mechanism [32]. We acknowledge two validity threats: differential history (category-specific market changes) and mechanism diffusion (generalized caution toward all labels) [19,33,34]. Testing consistency across these varied controls mitigates these threats.

### 2.2. Data Sources

We analyzed repeated cross-sectional data from four waves of the UK Food and You Survey (2012–2018) [35]. This biannual survey utilizes a nationally representative sample of adults in England, Wales, and Northern Ireland. The initial sample included 11,889 participants. Following the exclusion of individuals with missing data on sociodemographics (n=2892), readability perceptions (n=33), or consumption (n=16), the final analytical sample comprised 8948 individuals. Survey weights were applied to ensure national representativeness.

### 2.3. Outcome Variables

Participants reported consumption frequency for 14 food items. We focused on the target REMs (sandwiches, pre-cooked meats) and controls (dairy, fresh meat). The original eight-point frequency scale was collapsed into a three-point ordinal variable to ensure adequate cell counts and conceptual clarity: (1) Never; (2) Monthly aggregating once a fortnight, once a month, and less than once a month; and (3) Daily/Weekly aggregating frequencies from once or twice a week to at least once a day. This aggregation preserves the ordinal structure while producing statistically balanced groups.

### 2.4. Independent Variables

The primary independent variable is perceived MTL print size. In all waves, participants were asked: How easy do you find it to read the labelling on food products (e.g., ingredients, nutrition or storage information) in terms of the size of the print (using glasses or contact lenses if you wear them)? Responses were recorded on a 5-point Likert scale ranging from (1) Very difficult to (5) Very easy.

### 2.5. Control Variables

In addition to food consumption frequency and perceptions of MTL print size, we added several behavioural, location, and sociodemographic variables to the analysis, acknowledging their influence on REM consumption patterns [36]. This includes the presence of children under the age of 16 in the household, represented as a dichotomous variable (1 for presence, 0 for absence). Households with children tend to place a greater emphasis on nutrition, often leading to increased consumption of fruits and vegetables and reduced intake of ultra-processed foods due to heightened awareness of nutritional information [37,38]. Studies have demonstrated that family composition, in terms of the number of members and their relationships, is associated with lower dietary diversity [39]. In this study, household size is represented by an ordinal variable encompassing family members from 1 to 4+, and marital status is a categorical variable with binary coding (1 for married or living with partner; and 0 for single, widowed, divorced, separated, or other). Research indicates that religious affiliation can impact dietary choices, with studies suggesting that individuals who are religious tend to have less diverse diets or avoid certain foods [39]. Religion is included as a dummy variable to denote Christian affiliation and other religions, with no religion serving as the reference category.

Shopping responsibilities, which indicate whether an individual is responsible for (1) all or most of the food/grocery shopping, or (0) less than half of the purchases, capture the frequency with which individuals engage in food purchasing at home. Studies suggest a decline in food purchasing and cooking at home, coinciding with the increased availability of REM in markets. Consequently, individuals who are more actively involved in home food shopping and cooking are less inclined to consume ultra-processed foods [40].

Individual characteristics such as sex and household income were also included in the analysis [41]. Sex is a binary variable, with 1 representing male and 0 representing female. Age is an ordinal variable divided into seven distinct ranges: 16–24 years, 25–34 years, 35–44 years, 45–54 years, 55–64 years, 65–74 years, and 75 years and above. Household income was originally recorded in four categorical bands and dichotomized into a binary variable (1 = high, 0 = low) at the £26,000 threshold. This cutoff aligns with the original category boundaries and approximates the UK median household disposable income during the study period, effectively dividing the sample into lower- and higher-income segments.

The country of residence is also included as a dummy variable representing England, Wales, with Northern Ireland as the reference category. Another included geographical variable is the classification of urban (1) versus rural (0) areas. Recent research has indicated that diet-related concerns, such as food insecurity and obesity, are related to area-level deprivation in the UK [42]. Area-level deprivation indices were omitted because they were inconsistently available across the survey waves analyzed.

Although biological samples (e.g., urine, blood) were collected from a subset of participants in specific waves, these were not utilized in the current study. These samples remain a distinct avenue for future research, specifically to explore biomarkers related to the impact of MTL labelling on cardiovascular health.

### 2.6. Subgroup Analysis

To explore the salience-to-understanding pathway, we stratified analyses by consumer engagement [16]. Food safety concern was measured by the statement *I often worry about whether the food I have is safe to eat*. The original 5-point Likert scale was dichotomised into a high concern group (comprising those who selected definitely agree or tend to agree) and a low concern group (comprising all other respondents). Information-seeking behavior classified respondents as seekers or non-seekers based on their reported sources of food safety information. These moderators were selected as proxies for sustained label engagement, prioritizing persistent traits over acute experiences (e.g., food poisoning). Measures of visual acuity or validated health literacy scales were not available across all waves.

### 2.7. Statistical Analysis

We employed cumulative link proportional odds ordinal logistic regressions to investigate the associations between individuals’ perceptions of MTL print size and the consumption frequency of the target products (pre-packaged sandwiches and pre-cooked meat) and the non-equivalent dependent variables (dairy and fresh meat). The model is specified as follows:$$\log\left(\frac{P(Y\leq j)}{P(Y>j)}\right)=\alpha_{j}-\beta_{1}\mathrm{MTL}-\beta_{2}\left(\mathrm{MTL}\times \mathrm{Years}\right)-\gamma C$$
where *Y* denotes the ordinal result of the frequency of consumption for the specific food category analysed. *j* indexes the thresholds between the result categories. The threshold-specific intercepts $\alpha_{j}$ define the baseline log-odds of being below each threshold. $\beta_{1}$ represents the coefficient for the perception of MTL print size, and $\beta_{2}$ represents the interaction between readability and Years. C represents the vector of all control covariates (including the main effect of Years, sociodemographic factors, and behavioural characteristics), and γ is the corresponding vector of regression coefficients for these controls. A negative coefficient for $\beta_{1}$ indicates that participants who find the MTL print size easy to read are less likely to fall into a higher consumption frequency category.

Survey year was modeled as a categorical rather than continuous variable (with 2012 serving as the reference category). This specification enables detection of non-linear temporal trends and policy-specific period effects. Multicollinearity was assessed using Variance Inflation Factors (VIF) (see Appendix A and Appendix B). All predictors demonstrated low collinearity, with adjusted VIF values ranging from 1.02 to 1.65. These values are well below the conventional threshold of 5, confirming that the regression estimates are not biased by collinearity.

We tested the proportional odds assumption using the Brant test (see Appendix A and Appendix B). While the global test was statistically significant (p<0.001)—a common outcome in large samples driven by minor deviations in demographic covariates—the assumption of parallel regression held for the readability variable across all four product models (all p>0.50). Based on the results of this test, we utilized ordinal logistic regression models with proportional odds.

The analyses were carried out using the statistical software R version 4.4.2 [43], with a significance level set at 5%.

## 3. Results

### 3.1. Participant Characteristics

Table 1 details the sociodemographic characteristics of the 8948 participants, stratified by their consumption frequency of pre-packaged sandwiches, pre-cooked meat, dairy, and fresh meat. The demographic profiles of these consumer segments illustrate distinct consumption patterns across product categories. For sandwiches, high-frequency consumption was notably more prevalent among women (62.4%) compared to non-consumers (45.8%). This segment of frequent consumers was also significantly younger, with individuals in the 18–24 (19.9%) and 25–34 (23.8%) age brackets being highly represented, whereas the 65+ age group predominantly comprised non-consumers (27.8%). Furthermore, individuals without religious affiliation demonstrated a greater concentration among weekly consumers (41.9%) compared to those who never consumed these products (29.3%).

In contrast, the demographic profile of pre-cooked meat consumers presents distinct characteristics. Specifically, the gender distribution transitions from a predominantly male (55.0%) representation among non-consumers to a majority female (53.0%) composition among those who consume these products weekly. Contrary to sandwich consumption patterns, older age cohorts exhibit higher consumption rates, with individuals aged 65+ forming a significant proportion of both monthly (24.7%) and weekly (20.7%) consumers. Furthermore, religious affiliation demonstrates a divergent trend, as Christians constitute a substantial majority of weekly consumers (63.8%), whereas individuals identifying with other religions are disproportionately prevalent among non-consumers (22.8%). The consumption patterns for dairy and fresh meat also reveal unique demographic distributions. Dairy product consumption is nearly universal, with the predominant portion of the sample belonging to the weekly consumption category. This segment demonstrates a balanced gender representation (49.9% male) and comprises a considerable proportion of older consumers (21.0% aged 65+). The demographic profile for fresh meat consumption largely corresponds to that of pre-cooked meat, indicating a shift from a majority male (59.4%) non-consumer group to a majority female (53.1%) weekly consumer group, alongside a comparably high incidence of older adults (21.8%) within the weekly consumption cohort.

Across all four product categories, lower-income households consistently comprised a larger proportion of monthly and weekly consumer segments compared to non-consumers. Similarly, characteristics such as household size and marital status exhibited heterogeneous distributions across consumption levels, thereby highlighting the distinct demographic compositions of each consumer segment.

Table 2 indicates differentiated behavioral profiles across the product categories. Specifically, a discernible pattern regarding shopping responsibility is evident among sandwich consumers, where individuals with reduced responsibility (Sometimes/Never) exhibit a higher concentration within the frequent consumption segment (56.6%, weekly) compared to non-consumers (43.7%). This association, however, does not extend to other product categories. For pre-cooked meat, dairy, and fresh meat, the distribution of shopping responsibility demonstrates greater uniformity across all consumption frequencies.

Concerns pertaining to safe eating practices exhibited divergent relationships across the product categories. For pre-packaged sandwiches, a higher prevalence of individuals with significant safety concerns was observed among weekly (25.8%) consumers compared to monthly consumers (17.7%). Conversely, regarding pre-cooked meat, elevated safety concerns were most frequently noted among non-consumers (34.3%). This inverse correlation was even more pronounced for dairy products, where the highest level of concern was identified among individuals who never consumed the product (34.3%), while the lowest concern was evident among weekly consumers (20.8%). In contrast, for fresh meat, the degree of high concern remained notably consistent, at approximately 21%, across all consumption groups. Furthermore, information-seeking behavior emerged as a distinguishing factor exclusively for sandwich consumption, with active information seekers constituting a larger proportion of the weekly consumer segment (84.1%) relative to the non-consumer group (79.5%). However, for pre-cooked meat, dairy, and fresh meat, the proportion of information seekers remained stable at approximately 81–83% across all consumption levels, indicating that this behavior did not serve as a key differentiator for these particular products.

Temporal analysis of the survey year variable indicates shifts in the consumer base across all product categories. For sandwich consumption, the 2018 cohort demonstrated the highest proportion among monthly consumers (28.6%). A comparable trend was observed for pre-cooked meat, fresh meat, and dairy products, with the 2018 cohort also predominantly represented within their respective monthly consumption groups (30.2%, 33.7%, and 40.3%, respectively). Conversely, for these latter three categories, the 2018 cohort exhibited the lowest representation among weekly consumers, a pattern distinct from that seen for sandwiches.

### 3.2. Perceived MTL Print Size Readability

Figure 4 illustrates the perceived readability of MTL print size among diverse demographic and behavioral subgroups from 2012 to 2018. Where higher scores denote enhanced readability, the findings delineate significant disparities in food label perception across distinct subgroups. A pronounced and consistent age-related pattern is observed. Specifically, younger respondents (16–24 years) consistently reported the greatest ease of label comprehension, with their mean scores demonstrating an upward trajectory from 3.91 in 2012 to a peak of 4.23 in 2018. In contrast, middle-aged cohorts (45–54) experienced the most significant challenges in readability, indicated by a decline in scores from 2.88 to 2.58 during the corresponding timeframe. Older adults (65+), however, reported a moderate level of reading ease, characterized by stable mean scores approximating 3.2.

Household and socioeconomic factors also reveal distinct patterns. Respondents from high-income households (means around 3.5–3.6) consistently found labels easier to read than those from low-income households (means around 3.1–3.3). Similarly, those with no children at home reported significantly greater ease of reading (mean 3.75) than those with children (mean approximately 3.3). Geographically, residents in Wales reported finding labels easiest to read in 2012 (mean = 3.92), though this level later converged with England and Northern Ireland. Behavioral characteristics were also associated with perceived readability. Individuals who do not actively seek food information (mean 3.50 in 2012) reported greater ease of reading than active information seekers (mean 3.33 in 2012). Likewise, those with less shopping responsibility (Sometimes/Never) reported finding labels easier to read (mean approximately 3.5) than those with primary responsibility (mean around 3.4). Differences based on food safety concerns were less pronounced, with both high- and low-concern groups reporting similar levels of readability.

Regarding pre-packaged sandwiches, a distinct and consistent hierarchical pattern in perceived readability is evident across various consumption subgroups. Specifically, individuals categorized as never consumers consistently reported the highest mean readability scores, fluctuating between 3.78 in 2012 and 3.66 in 2018. Conversely, weekly consumers consistently indicated the lowest mean readability. In the context of pre-cooked meat, the observed pattern is less clearly delineated. Although monthly consumers generally reported the greatest ease of reading (mean 3.63 in 2012), the readability values across all three consumption segments were considerably similar and exhibited convergence over the study period, thus precluding the establishment of an enduring hierarchy. Dairy products revealed a hierarchical structure akin to that observed for sandwiches, albeit less pronounced: never consumers (mean 3.47 in 2012) and monthly consumers (mean 3.44 in 2012) consistently reported higher readability levels compared to the predominant weekly consumer subgroup. For fresh meat, the pattern is again less distinct and more closely resembles pre-cooked meat.

### 3.3. Perceived MTL Print Size Readability and Food Consumption Associations

Table 3 and Figure 5 present the findings derived from the ordinal logistic regressions conducted using non-equivalent dependent variables (NEDVs). Initial analysis revealed a statistically significant association exclusively concerning pre-packaged sandwiches (β=−0.10, p<0.05). Estimations elucidate a more intricate relationship when considering the interaction between perceived readability and the survey year. Specifically for pre-packaged sandwiches (Panel A), a notable dynamic relationship was discerned. During the baseline year of 2012, a statistically significant inverse relationship was observed (*OR* = 0.91, 95% CI [0.83, 0.99]), where in each one-unit increment in perceived readability correlated with an approximate 9% reduction in the odds of more frequent consumption. This negative relationship, reflected in the interaction terms, persisted through 2014 (−4%) and 2016 (−5%). This association underwent a transformation over the study period, evidenced by a significant positive interaction term for 2018 (Readability × 2018: *OR* = 1.15, 95% CI [1.02, 1.29]). Consequently, by 2018, the overall impact of readability had inverted (combined *OR* = 1.04), manifesting as a 4% increase in the odds of frequent consumption for each one-unit rise in perceived readability.

This dynamic association was not observed across the control outcome categories. Specifically, the analysis for pre-cooked meat (Panel B), dairy (Panel C), and fresh meat (Panel D) revealed no comparable temporal dynamic; neither the main association with readability nor its interactions with the survey year achieved statistical significance. This consistent pattern robustly indicates that the observed dynamic association is not a generalised consequence of label salience but rather an effect specific to the pre-packaged sandwich category. While this dynamic interaction was specific to this category, other significant temporal trends were observed across all models. Specifically, for pre-packaged sandwiches, the odds of frequent consumption in 2018 were reduced by 62% compared to those in 2012. In contrast, the odds of frequent consumption for pre-cooked meat demonstrated a significant elevation in subsequent years, increasing by 87% in 2016 and 73% in 2018 relative to the 2012 baseline. Furthermore, fresh meat consumption exhibited an even more pronounced increase, with the odds of frequent consumption in 2018 exceeding a two-fold rise.

### 3.4. Health Belief Associations

Table 4 and Table 5 show the findings from ordinal logistic regressions, stratified by health belief behavioral subgroups. The results confirms that the relationship between perceived MTL print size readability and consumption frequency is significantly influenced by product category, suggesting varied patterns of label engagement.

The analysis stratified by information-seeking behavior highlights a key divergence. Among non-information seekers, a significant negative correlation was observed between perceived readability and the frequency of pre-packaged sandwich consumption (*OR* = 0.81, 95% CI [0.69, 0.95]). In other words, each one-unit increment in perceived MTL print size readability correlated with an approximate 19% reduction in the odds of more frequent consumption. While the primary association did not achieve significance among information seekers, a statistically significant interaction with the year 2018 was exclusively observed within this group (*OR* = 1.15, 95% CI [1.01, 1.30]). This indicates that their relationship with readability has evolved, manifesting as a 7% increase in the odds of frequent consumption for each one-unit rise in perceived readability.

For the control pre-cooked meat, readability consistently showed no significant main or interaction effects in either group. This null finding is mirrored in the no-label falsification test, fresh meat. Notably, the dairy control analysis revealed a pattern analogous to that of sandwiches, where a significant main effect of readability was exclusively identified among non-information-seeking participants (*OR* = 0.65, 95% CI [0.46, 0.92]).

When stratifying by concerns regarding food safety, a comparable pattern was observed. Specifically for pre-packaged sandwiches, both the significant primary association with readability (*OR* = 0.90, 95% CI [0.82, 0.99]) and the notable interaction effect with the year 2018 (*OR* = 1.16, 95% CI [1.03, 1.31]) were observed solely among individuals expressing low levels of concern. In this subgroup, a one-unit increment in perceived MTL print size readability initially correlated with an approximate 10% reduction in the odds of more frequent consumption, a trend that subsequently evolved into a 4% increase by 2018. In the control categories of pre-cooked meat and dairy, readability did not emerge as a significant factor for either the high- or low-concern subgroups. Conversely, the no-label fresh meat model unexpectedly revealed a significant positive interaction with the year 2018 specifically within the high-concern group (*OR* = 1.40, 95% CI [1.08, 1.81]), despite the main effect of readability not reaching statistical significance for either subgroup.

### 3.5. Sensitivity Analysis

As a robustness check, we reframed the analysis using binary logistic regression, where consumption was classified as either Yes (any frequency) or No (never), while adjusting for the same control variables (see Appendix A and Appendix B). The results from this binary approach reveal a consistent directional association for pre-packaged sandwiches compared to the ordinal model. For this product, a statistically significant negative association emerged suggesting that each one-unit increment in perceived MTL print size readability correlated with an approximate 16% reduction in the odds of being a consumer. The findings for the control outcome categories aligned with those obtained from the primary ordinal model, indicating an absence of a statistically significant association between perceived readability and the propensity for consumption. While this robustness check confirms the product-specific nature of the findings, it is important to note that the binary classification of consumption is a simplification. This approach may mask more subtle relationships by not distinguishing between different levels of consumption frequency, such as monthly versus weekly, which were considered in the primary ordinal analysis.

We conducted a sensitivity analysis to address the possibility of reverse causality. Specifically, we modelled readability as the outcome and consumption frequency as the predictor, adjusting for the same covariates (see Appendix A and Appendix B). The results indicate that the direct association between these two variables was statistically significant for both pre-packaged sandwiches and pre-cooked meat. This significant reverse association underscores the importance of carefully considering the directional implications within the primary model. While our findings support strong reverse causality, the cross-sectional nature of the data means we cannot definitively establish the direction of the relationship. Future longitudinal studies are needed to clarify these dynamics.

While ready meal constitutes a distinct category within the UK Food and You Survey, its data availability was limited to 2016, thereby precluding the analysis of cross-sectional associations central to this research. For transparency, the ordinal logistic regression model for this specific category was included in the Appendix A and Appendix B. The resulting positive, non-statistically significant estimates corroborate the findings of our primary model for pre-cooked meat, although caution is required due to the small sample size.

To validate the assumption of linearity, we conducted a sensitivity analysis treating perceived print size readability as a categorical variable rather than a continuous one. The Target Product Sandwiches is the only category where perceived readability shows a significant negative association with consumption frequency (e.g., *OR* = 0.50 for Easy vs Very difficult), supporting the linear gradient observed in the main analysis (see Appendix A and Appendix B). In contrast, the coefficients for all three NEDVs (Pre-cooked Meat, Dairy, and Fresh Meat) are statistically non-significant (p>0.05) and fluctuate without a clear linear pattern. This confirms that the observed effect is specific to the target product and robust to the functional form of the variable.

We conducted an analysis comparing the primary full model against a parsimonious model. We specifically excluded religion, household size, urban/rural status, and the presence of children under 16 from the reduced model, as these variables demonstrated limited predictive power and statistically non-significant associations with the outcome in the full model. As shown in the Appendix A and Appendix B, the coefficient for perceived readability remained stable across all product categories. For the target product (sandwiches), the coefficient changed by less than 1%, confirming that the included demographic variables did not introduce confounding bias. Consequently, the full model was retained in the main analysis to strictly adjust for all potential sociodemographic variations.

Transportability is a critical component of external validity cheques [44,45], adjusting study results between two UK shopper populations by accounting for differences in baseline characteristic distributions [46]. In this study, our validation focused specifically on the sampling mechanism, accounting for the structural differences between the perception-based UK Food and You survey (approx. 3000 participants per wave) and the objective National Diet and Nutritional Survey (NDNS) (approx. 1000 participants per wave; 4-day consumption report).

We accomplished this by generalizing the Food and You findings to the NDNS target population, employing two generalized boosted models (GBM) with distinct sets of covariates. The first model included only demographic and socioeconomic covariates, while the second, which we refer to as the fully adjusted model in our main analysis, encompassed a broader set of variables. This approach strengthens external validity by mitigating selection biases inherent in the differing sampling designs. By demonstrating that the association holds even when the source sample is reweighted to match the demographic structure of the NDNS, we confirm that the findings are applicable to the broader policy-relevant population and are not idiosyncratic to the Food and You cohort.

Furthermore, to evaluate how well the transportability weights from our fully adjusted model balanced the covariate distributions, we performed two diagnostic checks. First, we created a table comparing the distributions of the shared covariates between the Food and You and NDNS samples. Second, we plotted the density of the estimated participation scores for both survey samples to visually assess their overlap and the success of the weighting adjustment.

## 4. Discussion

This study investigated the association between the perceived readability of MTL labelling print size and consumer choice, utilizing a non-equivalent dependent variable (NEDV) design to isolate content-specific effects [22,23]. Our findings reveal a relationship that is not merely product-specific but contingent on the interaction between the label content and the product consumption context. We observed a significant, dynamic association isolated exclusively to pre-packaged sandwiches, while finding persistent null effects for our other unhealthy labeled product (pre-cooked meat), our nutritionally advisable control (dairy), and our no-label falsification test (fresh meat).

This specific pattern provides strong evidence that the salience-to-understanding pathway functions as a warning mechanism. For pre-packaged sandwiches, readability was linked to a 9% decrease in the odds of frequent consumption in 2012 (OR=0.91), consistent with the intended deterrent effect of the red warning. This core finding was robust to transportability analysis [45], which validated the structural generalizability of this association to the broader dietary population represented in the NDNS. This strengthens confidence that the result is not an artifact of the specific UK Food and You Survey sample. Furthermore, the consistent absence of an effect for dairy and fresh meat supports the causal logic of the NEDV design: where the warning signal is absent—either due to green reinforcing codes or no label at all—increasing salience does not alter behavior.

The divergence between pre-packaged sandwiches and pre-cooked meat—both unhealthy, labeled REMs—warrants specific attention. We argue this stems from the distinct decision-making contexts characterizing ambivalent convenience, specifically the cognitive conflict between the immediate utility of a time-saving meal and the nutritional ambiguity regarding its healthfulness. Sandwiches are often time-pressured, discretionary, impulse-driven purchases where the health vs. convenience trade-off is salient at the moment of choice [3,47]. In this high-variance category, where the health halo of a product can be deceptive, a readable warning can tip the balance by resolving nutritional uncertainty.

In contrast, pre-cooked meat (e.g., ham, sliced chicken) often functions as a planned meal component or staple ingredient for evening meals [48]. For such habitual purchases, consumers may rely on top-down processing goals (e.g., I need sliced chicken for lunch) that override bottom-up visual cues like label print size [49]. Unlike sandwiches, pre-cooked meats may be viewed through a stable processed heuristic, rendering granular label details less influential. This suggests that label salience is most potent when the purchase decision is malleable and susceptible to immediate visual disruption.

Regarding the temporal dynamics, the reversal of the sandwich association by 2018—where readability became associated with increased consumption—likely reflects a combination of habituation and market adaptation. First, consistent with a time decay effect, the initial visual salience of the warning label may diminish over time as consumers habituate to the signal, reducing its disruptive power [50]. Second, empirical evidence suggests that manufacturers aggressively reformulated products following the MTL implementation to mitigate the stigma of ultra-processed foods [51,52]. Consequently, by 2018, a readable label may have signaled transparency or quality in a reformulated market, effectively neutralizing the warning effect. Theoretically, this implies a repurposing of the pathway; readability may have ceased to function as a deterrent and instead enhanced the salience of the product’s trustworthiness.

This relationship between attention and behavior also appears to be bidirectional. Our reverse causality sensitivity analysis indicated that frequent consumption predicts lower perceived readability. This suggests a potential feedback loop driven by habituation: as individuals consume these products more frequently, they may develop label blindness, engaging less with visual cues as purchasing becomes automatic. Thus, habitual consumers may perceive labels as less readable simply because they have stopped looking at them.

Finally, regarding consumer heterogeneity, our subgroup analysis aligns with the Health Belief Model. Readability predicted consumption primarily among less-engaged consumers (non-information seekers). For these individuals, who lack strong internal cues to seek nutrition info, external visual salience acts as a necessary call to action [53]. Conversely, highly engaged consumers likely employ directed attention strategies, seeking out nutritional data regardless of print size. This challenges the information overload hypothesis [18]; the issue for less-engaged groups is not too much data, but the perceptual accessibility of the primary warning signal.

Descriptively, our findings that older adults and lower-income groups report significantly lower perceived readability raise concerns about structural inequities. These perceptual disparities may signal underlying barriers related to visual health and health literacy [13,54,55]. If vulnerable groups physically struggle to discern the label, the policy fails them at the first hurdle of the communication pathway, exacerbating health inequalities.

Consequently, these findings suggest that a one-size-fits-all approach to labeling standards is insufficient. While increasing print size is a viable intervention, its effectiveness is bounded by packaging constraints. Policy should therefore consider adaptive standards: mandating larger, bolder warnings specifically for impulse categories where visual disruption is effective, while exploring alternative strategies—such as digital augmentation or simplified interpretative symbols—for categories where print size is constrained or less impactful.

### Limitations

This investigation offers valuable insights into real-world phenomena, though several limitations warrant consideration. We have categorized these into three key areas concerning study design and causal inference, measurement and bias, and statistical constraints.

Regarding study design and causal inference, the study employs a repeated cross-sectional design, pooling data from four waves. Although this precludes tracking individual behavioral trajectories, a strength of panel data, it suits our primary aim of modeling population-level changes in associations over time. By drawing fresh samples at each time point, it avoids attrition bias typical of long-term panel studies and maintains representativeness of the UK population throughout the policy period. However, the data structure inherently constrains the ability to draw definitive causal inferences.

Given the simultaneous nationwide implementation of the MTL policy in the UK, it was not feasible to distinguish distinct treatment groups. While our NEDV design mitigates general history threats, we acknowledge the potential for unobserved confounders that vary across food categories. For instance, unmeasured factors such as taste preferences or exposure to product-specific marketing campaigns could differentially influence the association for sandwiches compared to control products, potentially biasing the observed product-specific estimates. Consequently, the study is confined to reporting statistical associations rather than establishing causal effects. Furthermore, the observed reversal in the association for pre-packaged sandwiches may stem from concurrent, unmeasured events, such as broader public health campaigns, rather than solely reflecting individual-level adaptation to the MTL. Thus, population-level trends warrant cautious interpretation given the risk of an ecological fallacy.

With respect to measurement and bias, limitations exist regarding our key variables. Our central variable, perceived readability, is a self-reported proxy for the salience-to-understanding pathway, not a direct objective measure of comprehension. Furthermore, this measure pertains to food labelling in general, rather than specifically to the MTL panel, introducing potential measurement error if consumers struggle with ingredient lists but not the MTL. Future research should strengthen this by employing objective validations, such as eye-tracking or standardized legibility tests (e.g., ISO standards), to definitively disentangle visual acuity from perceived comprehension.

Similarly, we utilized food safety information seeking as a proxy for general engagement with nutritional information. We acknowledge that seeking information about safety (e.g., hygiene, poisoning) is conceptually distinct from seeking nutritional data. However, in the absence of a consistent nutritional seeking metric across all survey waves, this variable was selected as the most robust available indicator of active information-acquisition behavior. We assume a degree of overlap wherein vigilant consumers who actively verify safety information demonstrate a higher baseline of label engagement that likely extends to other informational cues, including nutrition.

The consumption data is also self-reported and subject to recall or social desirability biases. Crucially, these biases are likely differential across food categories, as consumers may be more prone to underreport the consumption of unhealthy items like sandwiches and pre-cooked meat compared to neutral staples like dairy, potentially attenuating the observed associations for the target products. Finally, while our study highlights the role of print size, it is methodologically challenging to fully disentangle its effect from interacting design elements, such as color contrast, layout, and typography, within an observational dataset.

Concerning statistical and data constraints, the dataset presented specific challenges, notably a significant portion of missing income data (>20% of the sample). We elected not to use multiple imputation, as the missingness was likely not random, and imputing such a large proportion of a primary socioeconomic covariate could introduce greater bias than listwise deletion in this context. Additionally, due to inconsistencies in how consumption frequency was measured across different food categories in the UK Food and You Survey, we could not extend this comparative analysis to other types of food products. Future research could address this limitation by utilizing household scanner data, which would allow for the analysis of a broader range of products and provide objective, granular measures of purchasing behavior.

Our measure of readability is subjective, representing a self-reported perception rather than an objective test of comprehension. Future research should strengthen this measurement strategy by employing objective validations, such as eye-tracking or standardized legibility tests (e.g., ISO standards), to definitively disentangle visual acuity from perceived comprehension and subjective readability. Finally, the content-dependent inference relies on the assumption that consumers interpret red MTLs as warnings; while supported by literature, individual interpretation of color codes varies.

## 5. Conclusions

The association between perceived MTL print size readability and consumption frequency is intricate and contingent not just on product type but on the label’s content and the product’s context. Our primary finding, strengthened by the use of non-equivalent dependent variables, indicates that the role of label salience is highly specific. We found a significant dynamic association isolated exclusively to pre-packaged sandwiches. This relationship reversed from a 9% decrease in the odds of frequent consumption in 2012 to a 4% net increase in the odds of frequent consumption per unit of readability by 2018. It is crucial to note that this 2018 result (OR=1.04) implies that higher readability was associated with more frequent consumption, a finding that contradicts the simple hypothesis that better readability of red warnings reduces intake. Conversely, a persistent null association was found for pre-cooked meat, our nutritionally advisable-labelled control (dairy), and our no-label falsification test (fresh meat). This pattern strongly supports our hypothesis that the salience-to-understanding pathway, as proxied by readability, functions as a conditional warning mechanism. Its effect appears specific to red health warnings on ambivalent convenience foods rather than a general effect of all labels. Furthermore, the large sample size provided sufficient statistical power to detect these subtle interaction effects, particularly within subgroups. The association for sandwiches was strongest among less-engaged consumer groups, supporting a model where readability acts as the primary behavioral cue for those lacking intrinsic motivation. Theoretically, our findings challenge the efficacy of a uniform approach to label-based interventions. Policy implications suggest that standardization alone may be insufficient. Instead, public health strategies should consider adaptive standards—mandating high-salience warnings for impulse categories where visual disruption is effective, while exploring alternative formats for categories where habituation or context neutralizes the warning signal.

## Figures and Tables

**Figure 1 nutrients-18-00197-f001:**
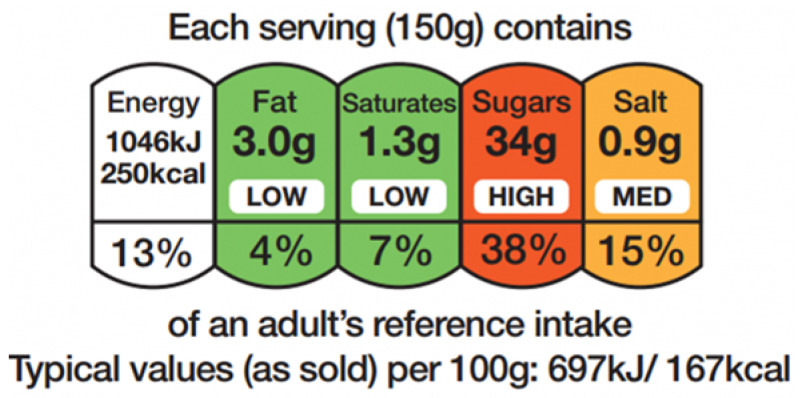
MTL labelling visual representation. MTL displays the percentages of the recommended daily nutrient intake as the numerical assessment of the product’s overall contribution for an average adult diet of 2000 calories.

**Figure 2 nutrients-18-00197-f002:**
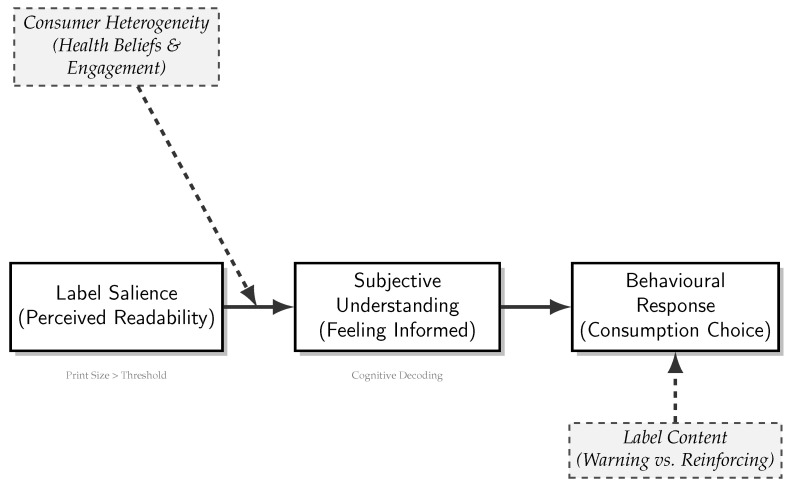
The salience-to-understanding conceptual framework. The model illustrates the hypothesized pathway where label readability (salience) facilitates subjective understanding, which in turn influences consumption behavior. This pathway is moderated by consumer heterogeneity (health beliefs and engagement) and the specific content of the label (warning vs. reinforcing signals).

**Figure 3 nutrients-18-00197-f003:**
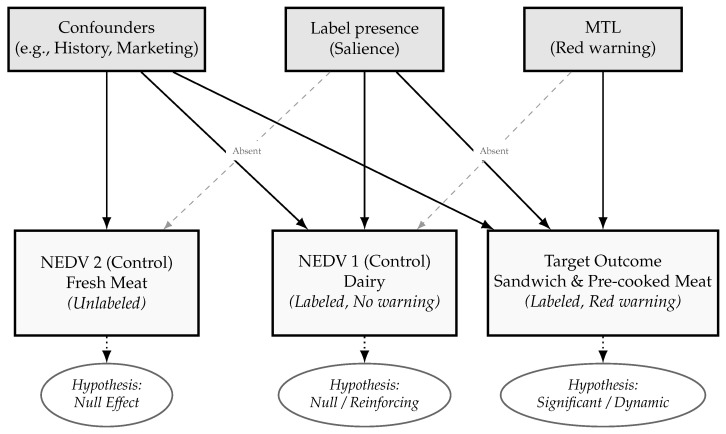
Conceptual logic of the Non-Equivalent Dependent Variable (NEDV) design. The design isolates the effect of the MTL warning mechanism by comparing the target outcome against controls that share general environmental confounders but lack the specific intervention mechanism. Solid lines represent exposure presence; dashed lines represent absence.

**Figure 4 nutrients-18-00197-f004:**
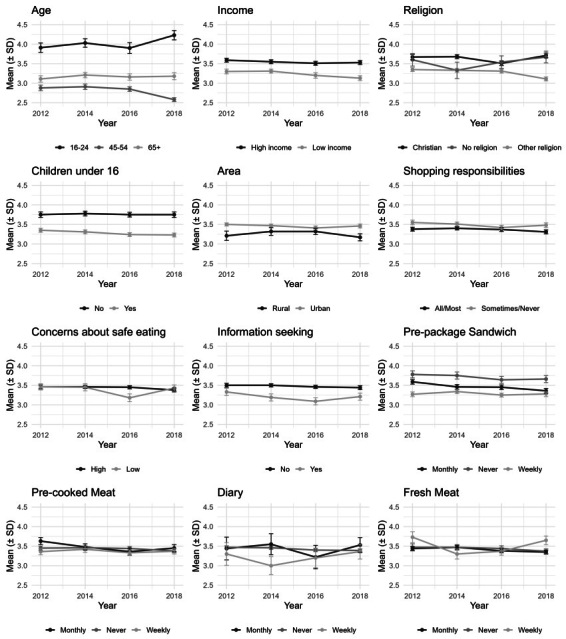
Mean perceived MTL print size readability from 2012 to 2018, stratified by sociodemographic characteristics, behavioral characteristics, food products. Higher scores denote enhanced readability.

**Figure 5 nutrients-18-00197-f005:**
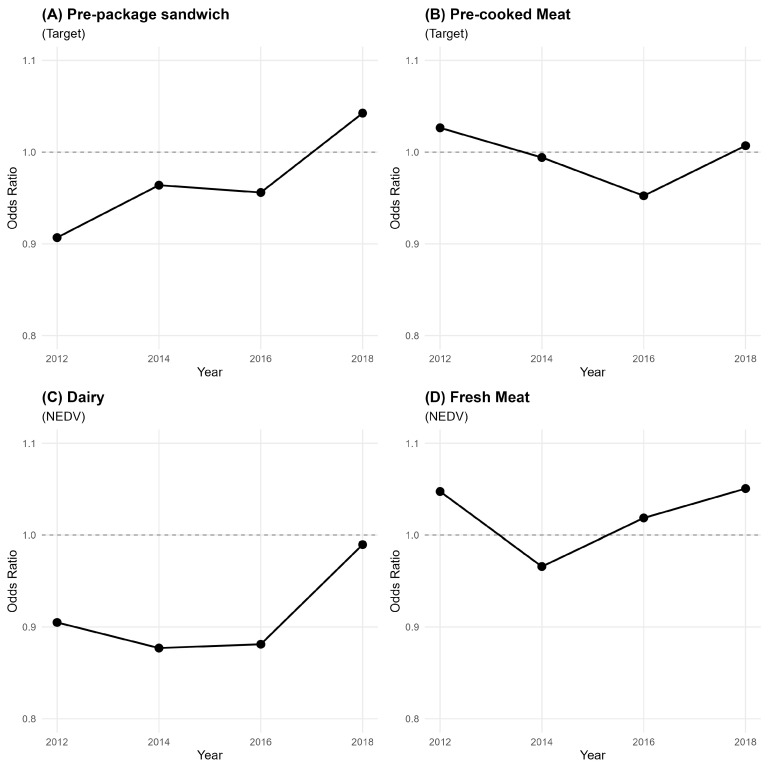
Perceived MTL print size readability and consumption frequency. Cross-sectional trends by product type.

**Table 1 nutrients-18-00197-t001:** Sociodemographic characteristics of participants by food consumption frequency (% within each consumption level).

	Pre-Packaged Sandwich			Pre-Cooked Meat			Dairy			Fresh Meat		
Category	Never	Monthly	Weekly	Never	Monthly	Weekly	Never	Monthly	Weekly	Never	Monthly	Weekly
n = 8948	4052	3221	1675	1615	2035	5298	246	203	8498	789	2197	5963
Sex												
Male	54.2	51.3	37.6	55.0	54.0	47.0	53.6	52.9	49.9	59.4	55.1	46.9
Female	45.8	48.7	62.4	45.0	46.0	53.0	46.4	47.1	50.1	40.6	44.9	53.1
Age												
18–24	7.7	10.9	19.9	14.6	9.0	10.9	8.1	14.7	11.1	12.5	10.7	11.1
25–34	14.2	18.1	23.8	19.3	18.4	16.4	18.1	21.1	17.3	17.4	17.8	17.2
35–44	15.5	18.1	19.7	18.2	15.4	17.6	18.5	15.7	17.2	19.1	17.7	16.8
45–54	19.1	20.3	20.0	20.8	17.8	20.1	19.3	20.6	19.7	20.2	21.0	19.2
55–64	15.7	14.3	8.2	11.2	14.7	14.2	19.2	12.5	13.7	11.7	14.2	13.9
65+	27.8	18.4	8.4	15.9	24.7	20.7	16.8	15.5	21.0	19.1	18.6	21.8
Religion												
Christian	63.8	58.0	48.7	42.1	59.5	63.8	55.5	54.0	59.1	42.4	57.4	61.6
Other religion	6.9	6.3	9.4	22.8	6.2	2.8	8.9	6.8	7.1	15.9	6.4	6.3
No religion	29.3	35.7	41.9	35.2	34.3	33.5	35.6	39.1	33.8	41.7	36.2	32.1
Marital status												
Married	39.0	37.5	51.7	44.3	41.2	39.6	46.3	51.8	40.4	45.4	42.4	39.6
Single	61.0	62.5	48.3	55.7	58.8	60.4	53.7	48.2	59.6	54.6	57.6	60.4
Household size												
1	19.5	14.2	17.9	16.5	20.1	16.5	25.3	24.7	16.9	20.5	19.6	16.0
2	38.7	39.3	28.7	30.8	41.4	37.3	30.8	29.7	37.4	29.5	36.2	38.3
3	17.4	18.3	24.2	21.2	17.9	18.7	17.6	18.3	19.0	21.8	17.7	19.1
4	24.4	28.2	29.2	31.6	20.7	27.5	26.3	27.3	26.7	28.3	26.5	26.6
Children at home												
Yes	72.6	68.8	67.7	64.4	75.8	70.0	75.3	69.3	70.2	65.6	69.8	71.1
No	27.4	31.2	32.3	35.6	24.2	30.0	24.7	30.7	29.8	34.4	30.2	28.9
Income												
Low	53.8	66.6	68.9	57.7	62.5	61.8	53.8	59.2	61.5	59.8	59.9	61.9
High	46.2	33.4	31.1	42.3	37.5	38.2	46.2	40.8	38.5	40.2	40.1	38.1
Area												
Urban	17.2	19.1	12.3	13.4	19.0	17.3	13.8	9.6	17.2	14.4	16.2	17.6
Rural	82.8	80.9	87.7	86.6	81.0	82.7	86.2	90.4	82.8	85.6	83.8	82.4
Country												
England	91.1	92.2	93.8	93.7	92.9	91.1	94.0	92.5	91.9	93.8	93.1	91.3
Wales	5.4	5.2	4.2	4.2	4.9	5.4	3.0	4.5	5.2	4.5	4.7	5.3
Northern Ireland	3.5	2.6	2.0	2.0	2.2	3.4	3.0	3.0	2.9	1.7	2.1	3.3

**Table 2 nutrients-18-00197-t002:** Behavioral characteristics of participants by food consumption frequency (% within each consumption level).

	Pre-Packaged Sandwich			Pre-Cooked Meat			Dairy			Fresh Meat		
Category	Never	Monthly	Weekly	Never	Monthly	Weekly	Never	Monthly	Weekly	Never	Monthly	Weekly
n = 8948	4052	3221	1675	1615	2035	5298	246	203	8498	789	2197	5963
Shopping												
responsability												
All/Most	56.3	50.9	43.4	51.7	56.2	50.4	58.9	53.6	51.7	53.1	54.3	50.9
Sometimes/Never	43.7	49.1	56.6	48.3	43.8	49.6	41.1	46.4	48.3	46.9	45.7	49.1
Concerns												
safe eating												
High	22.2	17.7	25.8	24.8	20.3	20.5	34.3	24.9	20.8	21.1	21.3	21.2
Low	77.8	82.3	74.2	75.2	79.7	79.5	65.7	75.1	79.2	78.9	78.7	78.8
Information												
seeking												
Yes	79.5	83.7	84.1	81.4	82.4	81.8	83.4	80.2	81.9	81.7	82.8	81.5
No	20.5	16.3	15.9	18.6	17.6	18.2	16.6	19.8	18.1	18.3	17.2	18.5
Year												
2012	26.0	22.1	22.0	20.7	18.8	26.8	15.3	13.9	24.3	18.1	17.5	27.0
2014	28.2	23.0	27.1	23.8	21.9	28.4	21.8	28.2	26.2	20.4	20.5	28.9
2016	25.8	26.4	24.8	27.7	29.1	24.0	26.2	17.6	26.0	32.1	28.2	24.1
2018	20.1	28.6	26.0	27.8	30.2	20.8	36.7	40.3	23.5	29.4	33.7	20.1

**Table 3 nutrients-18-00197-t003:** Ordinal logistic regression results: associations of Perceived MTL print size readability and food consumption frequency, including interactions (adjusted for all sociodemographic, behavioural, and temporal variables).

	Pre-Packaged Sandwich		Pre-Cooked Meat		Dairy		Fresh Meat	
Variable	β (SE)	OR [95% CI]	β (SE)	OR [95% CI]	β (SE)	OR [95% CI]	β (SE)	OR [95% CI]
Readability	−0.10 * (0.04)	0.91 [0.83, 0.99]	0.03 (0.03)	1.03 [0.96, 1.10]	−0.10 (0.11)	0.90 [0.72, 1.13]	0.05 (0.04)	1.05 [0.96, 1.14]
Year (vs. 2012)								
2014	−0.28 (0.23)	0.76 [0.49, 1.18]	0.15 (0.21)	1.16 [0.77, 1.75]	0.55 (0.60)	1.73 [0.53, 5.63]	0.27 (0.25)	1.31 [0.81, 2.12]
2016	−0.39 * (0.22)	0.68 [0.44, 1.05]	0.63 *** (0.19)	1.87 [1.30, 2.70]	0.52 (0.54)	1.68 [0.59, 4.82]	0.72 *** (0.20)	2.06 [1.39, 3.05]
2018	−0.96 *** (0.21)	0.38 [0.25, 0.58]	0.55 ** (0.19)	1.73 [1.20, 2.49]	0.78 (0.50)	2.19 [0.82, 5.84]	0.84 *** (0.20)	2.31 [1.57, 3.40]
Interactions								
Readability × 2014	0.06 (0.06)	1.06 [0.94, 1.20]	−0.03 (0.06)	0.97 [0.86, 1.09]	−0.03 (0.17)	0.97 [0.69, 1.35]	−0.08 (0.07)	0.92 [0.80, 1.06]
Readability × 2016	0.05 (0.06)	1.05 [0.93, 1.19]	−0.07 (0.05)	0.93 [0.84, 1.03]	−0.03 (0.16)	0.97 [0.72, 1.32]	−0.03 (0.06)	0.97 [0.87, 1.09]
Readability × 2018	0.14 * (0.06)	1.15 [1.02, 1.29]	−0.02 (0.05)	0.98 [0.89, 1.08]	0.09 (0.14)	1.09 [0.83, 1.44]	0.00 (0.06)	1.00 [0.89, 1.12]

Note: Table displays coefficients (β) with standard errors (SEs), odds ratios (ORs), and 95% confidence intervals (CIs). Significance levels: * p<0.05, ** p<0.01, *** p<0.001.

**Table 4 nutrients-18-00197-t004:** Ordinal logistic regression results: associations of perceived MTL print size readability and food consumption frequency by information-seeking subgroups (adjusted for all sociodemographic, behavioural, and temporal variables).

	Information Seekers		Non-Information Seekers	
Variable	β (SE)	OR [95% CI]	β (SE)	OR [95% CI]
Pre-packaged Sandwich				
Readability	−0.07 (0.05)	0.94 [0.85, 1.03]	−0.22 ** (0.08)	0.81 [0.69, 0.95]
Year (vs. 2012)				
2014	−0.17 (0.25)	0.84 [0.52, 1.36]	−0.47 (0.47)	0.63 [0.25, 1.57]
2016	−0.41 * (0.24)	0.66 [0.41, 1.06]	−0.28 (0.45)	0.75 [0.31, 1.81]
2018	−0.90 *** (0.23)	0.41 [0.26, 0.64]	−1.10 ** (0.43)	0.33 [0.14, 0.77]
Readability × Year				
2014 vs. 2012	0.05 (0.07)	1.05 [0.91, 1.20]	0.04 (0.13)	1.04 [0.80, 1.34]
2016 vs. 2012	0.08 (0.07)	1.08 [0.94, 1.24]	−0.08 (0.12)	0.93 [0.73, 1.17]
2018 vs. 2012	0.14 * (0.06)	1.15 [1.01, 1.30]	0.12 (0.12)	1.12 [0.88, 1.43]
Pre-cooked Meat				
Readability	0.06 (0.05)	1.06 [0.97, 1.16]	−0.06 (0.08)	0.94 [0.79, 1.11]
Year (vs. 2012)				
2014	0.24 (0.26)	1.28 [0.77, 2.12]	−0.36 (0.46)	0.70 [0.28, 1.73]
2016	0.66 ** (0.24)	1.93 [1.21, 3.09]	0.83 * (0.40)	2.29 [1.04, 5.04]
2018	0.76 *** (0.24)	2.15 [1.35, 3.42]	0.51 (0.37)	1.67 [0.80, 3.46]
Readability × Year				
2014 vs. 2012	−0.06 (0.07)	0.95 [0.82, 1.09]	0.12 (0.13)	1.13 [0.87, 1.46]
2016 vs. 2012	−0.06 (0.07)	0.94 [0.82, 1.07]	−0.16 (0.13)	0.85 [0.67, 1.09]
2018 vs. 2012	−0.05 (0.07)	0.95 [0.84, 1.08]	−0.06 (0.11)	0.95 [0.76, 1.18]
Dairy				
Readability	−0.02 (0.13)	0.98 [0.75, 1.27]	−0.44 * (0.18)	0.65 [0.46, 0.92]
Year (vs. 2012)				
2014	0.61 (0.73)	1.85 [0.44, 7.74]	0.55 (0.81)	1.74 [0.36, 8.44]
2016	0.71 (0.64)	2.03 [0.58, 7.12]	−0.02 (0.90)	0.98 [0.17, 5.67]
2018	1.22 * (0.57)	3.40 [1.11, 10.4]	−0.82 (0.81)	0.44 [0.09, 2.16]
Readability × Year				
2014 vs. 2012	−0.06 (0.20)	0.94 [0.63, 1.39]	0.06 (0.25)	1.06 [0.65, 1.73]
2016 vs. 2012	−0.07 (0.18)	0.93 [0.65, 1.33]	0.15 (0.29)	1.16 [0.66, 2.05]
2018 vs. 2012	0.00 (0.16)	1.00 [0.73, 1.37]	0.45 * (0.25)	1.57 [0.97, 2.55]
Fresh Meat				
Readability	0.09 (0.06)	1.09 [0.98, 1.22]	−0.08 (0.10)	0.92 [0.75, 1.13]
Year (vs. 2012)				
2014	0.53 (0.28)	1.69 [0.98, 2.92]	−0.19 (0.52)	0.82 [0.30, 2.26]
2016	0.90 *** (0.26)	2.45 [1.47, 4.09]	0.44 (0.47)	1.55 [0.62, 3.87]
2018	1.09 *** (0.25)	2.97 [1.82, 4.84]	0.30 (0.44)	1.35 [0.57, 3.20]
Readability × Year				
2014 vs. 2012	−0.13 (0.08)	0.88 [0.75, 1.03]	0.03 (0.14)	1.03 [0.79, 1.36]
2016 vs. 2012	−0.06 (0.07)	0.94 [0.82, 1.09]	0.03 (0.14)	1.03 [0.78, 1.35]
2018 vs. 2012	−0.05 (0.07)	0.95 [0.83, 1.09]	0.14 (0.13)	1.15 [0.89, 1.48]

Note: Table displays coefficients (β), standard errors (SEs), odds ratios (ORs), and 95% confidence intervals (CIs). Significance levels: * p<0.05, ** p<0.01, *** p<0.001.

**Table 5 nutrients-18-00197-t005:** Ordinal logistic regression results: associations of perceived MTL print size readability and food consumption frequency by concern subgroups (adjusted for all sociodemographic, behavioural, and temporal variables).

	High Concern		Low Concern	
Variable	β (SE)	OR [95% CI]	β (SE)	OR [95% CI]
Pre-packaged Sandwich				
Readability	−0.09 (0.10)	0.92 [0.76, 1.12]	−0.11 * (0.05)	0.90 [0.82, 0.99]
Year (vs. 2012)				
2014	−0.48 (0.47)	0.62 [0.24, 1.55]	−0.17 (0.24)	0.84 [0.53, 1.34]
2016	−0.66 (0.48)	0.52 [0.20, 1.32]	−0.29 (0.24)	0.75 [0.47, 1.20]
2018	−0.84 (0.46)	0.43 [0.18, 1.06]	−1.00 *** (0.24)	0.37 [0.23, 0.58]
Readability × Year				
2014 vs. 2012	0.12 (0.13)	1.13 [0.87, 1.46]	0.03 (0.07)	1.03 [0.91, 1.17]
2016 vs. 2012	0.11 (0.13)	1.12 [0.86, 1.44]	0.03 (0.07)	1.03 [0.90, 1.18]
2018 vs. 2012	0.11 (0.13)	1.12 [0.87, 1.44]	0.15 * (0.06)	1.16 [1.03, 1.31]
Pre-cooked Meat				
Readability	0.04 (0.09)	1.04 [0.88, 1.23]	0.02 (0.05)	1.02 [0.93, 1.12]
Year (vs. 2012)				
2014	0.05 (0.48)	1.06 [0.41, 2.73]	0.18 (0.26)	1.19 [0.71, 2.00]
2016	0.85 * (0.43)	2.35 [1.01, 5.46]	0.63 ** (0.23)	1.87 [1.20, 2.92]
2018	0.69 (0.45)	2.00 [0.83, 4.81]	0.64 ** (0.22)	1.90 [1.24, 2.92]
Readability × Year				
2014 vs. 2012	0.05 (0.13)	1.05 [0.81, 1.36]	−0.05 (0.07)	0.95 [0.83, 1.09]
2016 vs. 2012	−0.17 (0.12)	0.85 [0.67, 1.07]	−0.05 (0.06)	0.95 [0.84, 1.08]
2018 vs. 2012	0.01 (0.13)	1.01 [0.79, 1.30]	−0.05 (0.06)	0.96 [0.85, 1.08]
Dairy				
Readability	−0.04 (0.17)	0.96 [0.69, 1.34]	−0.12 (0.15)	0.89 [0.66, 1.19]
Year (vs. 2012)				
2014	1.32 (0.90)	3.76 [0.64, 22.0]	0.22 (0.74)	1.25 [0.29, 5.37]
2016	0.29 (0.89)	1.34 [0.24, 7.61]	0.59 (0.69)	1.80 [0.47, 6.94]
2018	1.24 * (0.71)	3.46 [0.85, 14.0]	0.63 (0.65)	1.87 [0.52, 6.70]
Readability × Year				
2014 vs. 2012	−0.25 (0.26)	0.78 [0.47, 1.29]	0.05 (0.21)	1.05 [0.70, 1.59]
2016 vs. 2012	0.09 (0.26)	1.10 [0.66, 1.82]	−0.06 (0.19)	0.94 [0.65, 1.37]
2018 vs. 2012	−0.04 (0.22)	0.96 [0.63, 1.47]	0.15 (0.18)	1.16 [0.81, 1.65]
Fresh Meat				
Readability	−0.14 (0.10)	0.87 [0.72, 1.06]	0.11 (0.05)	1.11 [0.99, 1.24]
Year (vs. 2012)				
2014	−0.47 (0.51)	0.63 [0.23, 1.70]	0.63 * (0.28)	1.88 [1.09, 3.24]
2016	0.07 (0.43)	1.07 [0.47, 2.47]	1.01 *** (0.25)	2.74 [1.68, 4.45]
2018	−0.34 (0.45)	0.71 [0.29, 1.71]	1.23 *** (0.24)	3.44 [2.16, 5.46]
Readability × Year				
2014 vs. 2012	0.16 (0.15)	1.17 [0.87, 1.57]	−0.17 * (0.08)	0.84 [0.72, 0.99]
2016 vs. 2012	0.14 (0.12)	1.15 [0.90, 1.47]	−0.09 (0.07)	0.92 [0.80, 1.05]
2018 vs. 2012	0.34 ** (0.13)	1.40 [1.08, 1.81]	−0.09 (0.07)	0.91 [0.80, 1.04]

Note: Table displays coefficients (β), standard errors (SEs), odds ratios (ORs), and 95% confidence intervals (CIs). Significance levels: * p<0.05, ** p<0.01, *** p<0.001.

## Data Availability

Publicly available datasets were analyzed in this study. The data from the Food and You Survey can be found from the UK Data Service at https://beta.ukdataservice.ac.uk/datacatalogue/series/series?id=2000053 (accessed on 3 November 2023). Additional data are available on the UK Food Standards Agency (FSA) website at https://data.food.gov.uk/catalog/datasets/3f3ad1b7-8cf3-444b-abbf-f784ea4551e1 (accessed on 28 October 2024). Data for the National Diet and Nutrition Survey can be found at https://datacatalogue.ukdataservice.ac.uk/studies/study/6533#details (accessed on 9 April 2024). Further guidelines and calorie reduction targets are available from Public Health England via https://www.gov.uk/government/organisations/public-health-england (accessed on 28 October 2024).

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
