# Peer review of "Food Label Readability and Consumption Frequency: Isolating Content-Specific Effects via a Non-Equivalent Dependent Variable Design"

_nutrients, 2026, doi:10.3390/nu18020197_

Round 1

Reviewer 1 Report

Comments and Suggestions for Authors

Thank you for the opportunity to review this manuscript, which examines when and for whom food label readability influences consumption behavior. The article seeks to disentangle content-specific from general readability effects by applying a non-equivalent dependent variable design to UK survey data. The topic is timely and relevant to ongoing debates about the contextual effectiveness of front-of-pack labeling systems. The study is dense, and I think that corrections will further strengthen the validity of the findings and the presentation of ideas.

RECOMMENDATIONS FOR THE TITLE AND ABSTRACT

  • You may consider slightly tightening it by removing the question format and using a more declarative structure, which is often preferred in academic journals. Additionally, while the phrase “content-specific effects on food choice” accurately captures the article’s contribution, you may clarify whether “food choice” refers specifically to self-reported consumption frequency, since this construct is narrower than general food choice behavior.
  • The objective is well stated, but the narrative may benefit from briefly clarifying why print size readability is a theoretically important dimension of label salience, especially relative to alternative salience cues (e.g., color, placement).
  • The methods section is appropriately detailed. Still, it could improve transparency by clarifying whether the cumulative link ordinal logistic regressions were adjusted for sociodemographic covariates, health behaviors, or survey wave fixed effects.
  • While the results are clearly summarized, the description of the dynamic relationship (from −9% to +4%) would benefit from briefly noting whether these effect sizes represent odds ratios per unit increase in self-reported readability or another standardization.
  • You may consider adding language that acknowledges the statistical power for detecting minor effects, especially across subgroups.
  • The conclusion effectively emphasizes the contextual nature of label effects, but could be strengthened by explicitly linking back to policy implications (e.g., whether standardization should be revisited or tailored).
  • The abstract is generally clear, though the Results section contains long sentences that may read more smoothly if divided, particularly the sentence beginning “The findings reveal…”.

CORRECTIONS PROPOSED TO THE INTRODUCTION

  1. The Introduction provides a comprehensive and well-structured overview of the public health context, but its length and density may obscure the central narrative. Consider streamlining the epidemiological background or relocating some descriptive statistics to a supplementary section to maintain conceptual focus. Additionally, while the distinction between ultra-processed foods broadly and ready-to-eat meals (REM) is explicitly sound, the argument would benefit from a more explicit explanation of why REMs serve as an ideal case for studying label readability, particularly relative to other ultra-processed categories.
  2. You may strengthen the causal logic by articulating why print-size readability is theoretically separable from other visual-salience features, and how consumers’ perceptual thresholds relate to policy-mandated minimum font sizes.
  3. You might emphasize more clearly how the proposed mechanism (readability → subjective understanding → behavior) synthesizes existing strands of the literature.
  4. Readers may benefit from an explicit conceptual model outlining the hypothesized pathway and its empirically testable components.
  5. The paragraph discussing health beliefs and motivation introduces substantial heterogeneity. Still, it may be helpful to tie this back to subgroup expectations earlier in the Introduction to prefigure the empirical strategy.
  6. The methodological challenge regarding the valence of label content is a critical contribution; nonetheless, the explanation of opposing effects (warning vs. reinforcement) could be more concise to preserve attention on the identification strategy.
  7. The justification for the non-equivalent dependent variable (NEDV) design is strong, though adding a brief contrast with alternative strategies (e.g., instrumental variables, discontinuities in implementation, eye-tracking) could strengthen the rationale.
  8. The justification for dairy as “healthy-labeled” could be tightened, particularly given public debates on dairy’s nutritional profile.
  9. You may consider numbering the hypotheses explicitly (e.g., H1, H2) to improve readability and later alignment with the Results section.
  10. The section is generally well written but overly long, with multiple sentences exceeding 35–40 words. Consider reducing internal parenthetical citations, splitting long sentences, and correcting minor agreement issues (e.g., “REM meals have emerged” → “have emerged”; “far exceeds” → “far exceed”).

CORRECTIONS PROPOSED TO THE MATERIALS AND METHODS

  1. You may strengthen credibility by clarifying how the chosen NEDVs (dairy and fresh meat) satisfy the assumptions of non-responsiveness to the MTL warning mechanism while remaining responsive to broader confounders.
  2. Consider adding a concise diagram or conceptual figure illustrating the NEDV logic, particularly since many readers may be unfamiliar with the method in nutritional epidemiology.
  3. While the section cites the limitations of conventional control groups, it would help to more explicitly articulate alternative quasi-experimental designs considered and why they were deemed unsuitable.
  4. The phrasing “This strengthens the evidence that the change… was not attributable to common threats” is appropriate. Still, reviewers may expect a discussion of potential violations of the NEDV assumption (e.g., marketing campaigns affecting dairy and REM differently).
  5. Some sentences contain redundancies (“consequently… therefore”), and others are overly long, diminishing readability; consider tightening for conciseness.
  6. The description of the Food and You Survey is clear; however, you might briefly justify why repeated cross-sections are adequate for capturing dynamic associations, highlighting any limitations relative to panel data.
  7. Consider reporting whether excluded respondents differ systematically from the analytic sample.
  8. It may be helpful to clarify whether survey weights were applied to ensure national representativeness across waves.
  9. Since biological samples were collected “in some instances,” consider stating whether these play any role in the current study.
  10. The decision to collapse the 8-category frequency scale into three ordinal categories is sensible. Still, you may wish to provide evidence that this aggregation does not mask meaningful variation or distort proportional odds assumptions.
  11. Consider noting whether sensitivity analyses were performed using the full 8-point scale to verify robustness.
  12. It would be helpful to specify the distribution of responses across the three categories to give readers a sense of balance.
  13. The operationalization of perceived print size is appropriate. However, you might acknowledge that the question refers to all label information (not specifically MTL), which could introduce measurement error if other label components drive perceptions.
  14. Consider explaining whether the Likert-scale variable was treated as continuous or tested for non-linearity.
  15. The temporal stability of the measurement across waves could also be briefly addressed (e.g., wording consistency).
  16. The inclusion of extensive sociodemographic and behavioral controls is justified; however, their theoretical link to label readability (as opposed to consumption frequency) could be clarified.
  17. Some variables (e.g., religion, household size) appear only weakly connected to label perception; consider whether all controls are necessary for parsimonious modeling.
  18. The rationale for dichotomizing income at £26,000 may require explanation, as this cutoff may not match conventional SES thresholds.
  19. The description of the urban/rural context could note whether area-level deprivation indices were considered or available.
  20. The text occasionally repeats conceptual justifications; condensing explanations for each control variable could improve readability.
  21. The selection of subgroups based on safety concerns and information-seeking aligns well with the theoretical pathway; however, the dichotomization of Likert scales may lead to information loss—consider acknowledging this limitation.
  22. It may be helpful to clarify whether subgroup interactions were pre-registered or exploratory.
  23. Guide why these particular moderators were prioritized over others (e.g., health literacy, visual acuity).
  24. The model specification is appropriate, but the notation could be streamlined; for instance, defining the full vector of controls rather than mixing β₃ and Σβâ‚–Xâ‚– would improve clarity.
  25. Consider explaining whether proportional odds assumptions were evaluated separately for each product category, given potential heterogeneity across outcomes.
  26. It would be helpful to explicitly state how “Years” was coded (continuous, categorical, centered).
  27. The statement about VIF and parallel odds assumptions is clear; providing ranges or summary statistics in-text would increase transparency.

CORRECTIONS PROPOSED TO THE RESULTS

The results section is clearly and coherently written, presenting the findings in a structured and logical manner. The narrative effectively guides the reader through the key associations, subgroup patterns, and comparative analyses, making the results both accessible and compelling. However, key modifications are required following the proposed corrections to the Methods section.

CORRECTIONS PROPOSED TO THE DISCUSSION

  1. The Discussion provides a strong summary of the main findings; however, the narrative could be streamlined by clearly separating the interpretation of effects, mechanisms, and implications, as these themes are currently interwoven and may obscure key arguments.
  2. The explanation of the content-dependent mechanism is well motivated. Still, the discussion might benefit from a more explicit articulation of why pre-packaged sandwiches—rather than pre-cooked meat—exhibited a significant effect despite both being REM products with red warnings.
  3. The interpretation of the reversal of effects in 2018 is plausible, yet largely speculative. Consider framing these explanations more carefully or referencing additional empirical evidence to support changes in marketing strategies or reformulation trends.
  4. The Discussion is generally readable but overly long, with several dense paragraphs. Dividing these and reducing overlap among themes would improve clarity.
  5. The caloric reduction example (52 vs. 12 sandwiches) is illustrative but may distract from the theoretical argument; consider integrating it more tightly with your conceptual model or moving it to a footnote.
  6. The transportability analysis is mentioned as validation. Still, the discussion could clarify which aspect of transportability (sample, mechanism, or structural assumptions) was tested and how this strengthens external validity.
  7. The explanation for null findings in dairy and fresh meat is coherent, but a brief reflection on why pre-cooked meat did not behave symmetrically to sandwiches would enhance interpretability.
  8. The subgroup findings are clearly presented, though it would help to contextualize these within existing theories of selective attention, health motivation, or label engagement, rather than referencing them primarily descriptively.
  9. The section on demographic correlates of readability is helpful, but consider exploring whether these perceptual disparities might signal structural inequities (e.g., visual health, literacy) relevant to labeling policy.
  10. The argument that MTL effectiveness depends more on consumer engagement than on the dissemination of information aligns with nudging theory; however, adding a contrast with competing frameworks (e.g., information overload, reactance) could deepen the theoretical contribution.
  11. The policy implications are well articulated, though you may consider discussing practical challenges—such as packaging constraints, industry compliance, or trade regulations—associated with modifying print size.
  12. The limitations section is comprehensive, though it may benefit from sharper organization: grouping limitations into categories (data, measurement, causal inference, statistical constraints) would enhance readability.
  13. The acknowledgement that causal inference is limited is appropriate; however, you may wish to more directly address whether unobserved confounders could bias the observed associations, particularly those that vary across food categories.
  14. The distinction between subjective readability and objective understanding is essential; consider referencing validated readability scales or alternative measurement approaches to strengthen future research recommendations.
  15. The discussion of recall and social desirability bias is appropriate, but you may add whether these biases are likely to be differential across food categories.
  16. The limitation regarding interacting design elements (color, layout, typography) is well stated; this presents an opportunity to briefly highlight the methodological challenges of isolating print size effects in observational data.
  17. The treatment of missing income data is transparent; however, readers may expect a brief justification for not applying multiple imputation despite >20% missingness (e.g., violation of MAR assumptions).
  18. The inability to extend analyses to other food categories is a valid limitation; consider noting whether future work could use alternative datasets (e.g., household scanner data) to address this.

RECOMMENDATIONS FOR IMPROVING THE CONCLUSION SECTION

  • The paragraph would benefit from improved punctuation—particularly the inclusion of commas in compound sentences—and from breaking the text into two shorter paragraphs to enhance readability.
  • The Conclusions effectively summarize the central findings; however, they might benefit from a more precise articulation of the conceptual contribution—specifically, how the study advances theory on salience, subjective understanding, and behavioral responses to labeling.
  • The statement that the association is “intricate and contingent” is accurate but somewhat broad; consider specifying which dimensions of contingency (content, context, consumer engagement) are most theoretically meaningful.
  • The emphasis on the dynamic reversal of effects is essential. Yet, the Conclusions could be strengthened by highlighting what this dynamic implies for label policy—e.g., the risk that label effects may attenuate or invert over time.
  • The discussion of null effects for comparison categories is appropriate. Still, you may wish to clarify how these null findings support the causal logic of the NEDV design and strengthen internal validity.
  • The claim that readability operates as a warning mechanism specifically for “ambivalent convenience foods” is compelling; however, adding a short clarifying clause on what characterizes “ambivalence” (nutritional ambiguity? competing attributes?) would bolster conceptual precision.
  • The section nicely identifies less-engaged consumers as the group most responsive to readability, but the conclusion could more explicitly address the implications for segmentation in public health policy.
  • While the Conclusions rightly challenge uniform labeling approaches, you may consider briefly noting potential alternative strategies (e.g., targeted messaging, adaptive design standards) to give policymakers clearer direction.

Thank you again for the opportunity to review this work; I appreciate the manuscript’s clear focus and potential contribution to the literature on labeling effectiveness.

Author Response

Dear editor and reviewers

Response to Reviewer 1:

Title and Abstract 

Comment 1. Title tightening and clarification: You may consider slightly tightening it by removing the question format and using a more declarative structure... Additionally, while the phrase “content-specific effects on food choice” accurately captures the article’s contribution, you may clarify whether “food choice” refers specifically to self-reported consumption frequency... 

Response: We have revised the title to a more declarative structure and clarified the outcome variable. The new title is: "Food Label Readability and Consumption Frequency: Isolating Content-Specific Effects via a Non-Equivalent Dependent Variable Design." 

Comment 2. Abstract - Theoretical importance of readability: The objective is well stated, but the narrative may benefit from briefly clarifying why print size readability is a theoretically important dimension of label salience... 

Response: We have amended the abstract's objective to define readability as a "theoretical structural gatekeeper for visual salience," clarifying its foundational role before other salience cues (like color) can be effective. 

Comment 3. Abstract - Methodological adjustments: ...clarifying whether the cumulative link ordinal logistic regressions were adjusted for sociodemographic covariates, health behaviors, or survey wave fixed effects. 

Response: The abstract now explicitly states: "Models were adjusted for sociodemographic covariates, health behaviors, and survey wave fixed effects." 

Comment 4. Abstract - Effect size clarification: ...description of the dynamic relationship (from −9% to +4%) would benefit from briefly noting whether these effect sizes represent odds ratios per unit increase in self-reported readability... 

Response: We have clarified this in the abstract, stating that the effect sizes represent the change in the odds of frequent consumption "per unit increase in readability," specifying the direction (decrease/increase) and providing the corresponding Odds Ratios ($OR=0.91$ and $OR=1.04$, respectively). 

Comment 5. Abstract - Statistical power: You may consider adding language that acknowledges the statistical power for detecting minor effects... 

Response: While we have added a sentence on statistical power to the Conclusion section of the main text, we did not have sufficient space to include it in the abstract without exceeding word count limits. We hope the inclusion in the main text is sufficient. 

Comment 6. Abstract - Policy implications: The conclusion... could be strengthened by explicitly linking back to policy implications (e.g., whether standardization should be revisited or tailored). 

Response: The abstract's conclusion now explicitly states: "...highlighting the need for adaptive rather than uniform policy standards." 

Comment 7. Abstract - Readability: ...the Results section contains long sentences that may read more smoothly if divided... 

Response: We have divided long sentences in the abstract's Results section to improve readability. 

Introduction 

Comment 8. Causal logic of print-size readability: You may strengthen the causal logic by articulating why print-size readability is theoretically separable from other visual-salience features... 

Response: We have added a paragraph to the Introduction explicitly defining and distinguishing between salience (initial capture), readability (decoding text), objective understanding (accuracy), and subjective understanding (feeling informed). We argue that readability functions as the "structural gatekeeper" that is a necessary condition for cognitive decoding, separate from simple attentional capture by color. 

Comment 9. Mechanism synthesis: You might emphasize more clearly how the proposed mechanism (readability → subjective understanding → behavior) synthesizes existing strands of the literature. 

Response: The revised Introduction more clearly articulates the "salience-to-understanding mechanism," explaining how a label must traverse a pathway from visual detection to subjective understanding to influence behavior, citing relevant theoretical frameworks and models. 

Comment 10. Conceptual model: Readers may benefit from an explicit conceptual model outlining the hypothesized pathway... 

Response: Figure 2 in the manuscript is a conceptual diagram outlining the hypothesized "salience-to-understanding pathway," including moderators like health beliefs. 

Comment 11. Subgroup heterogeneity: ...it may be helpful to tie this back to subgroup expectations earlier in the Introduction to prefigure the empirical strategy. 

Response: We have integrated the discussion of consumer heterogeneity and the Health Belief Model into the description of our conceptual framework in the Introduction, explicitly stating our expectation for substantial heterogeneity across consumer subgroups. 

Comment 12. Valance of label content: ...the explanation of opposing effects (warning vs. reinforcement) could be more concise... 

Response: We have streamlined the explanation of opposing effects for unhealthy versus healthier products to be more concise while maintaining clarity on the identification strategy. 

Comment 13. NEDV justification & alternatives: The justification for the non-equivalent dependent variable (NEDV) design is strong, though adding a brief contrast with alternative strategies... could strengthen the rationale. 

Response: We have added a paragraph to the "Study design" section explicitly detailing why alternative designs like Interrupted Time Series (ITS) and Difference-in-Differences (DiD) were unsuitable due to data resolution and the lack of a control group, respectively, thus justifying the NEDV approach. 

Comment 14. Dairy as "healthy-labeled": The justification for dairy as “healthy-labeled” could be tightened, particularly given public debates on dairy’s nutritional profile. 

Response: In the "Study design" section, we clarified that dairy products typically feature green/amber codes and thus serve a reinforcing role, while acknowledging that they are not universally "healthy." We include the concept of nutritionally advisable products. 

Comment 15. Hypothesis numbering: You may consider numbering the hypotheses explicitly... 

Response: The hypotheses at the end of the Introduction are now explicitly numbered (H1, H2, H3). 

Comment 16. Introduction length and focus: The Introduction provides a comprehensive... overview... but its length and density may obscure the central narrative. Consider streamlining... 

Response: We have significantly streamlined the Introduction, reducing readability-diminishing sentences and parenthetical citations. We have focused the narrative on the core research question and the unique suitability of REMs for this study. 

Comment 17. Writing style: The section is generally well written but overly long, with multiple sentences exceeding 35–40 words. Consider reducing... 

Response: We have conducted a thorough review of the entire manuscript to improve readability, splitting long sentences, reducing redundancies, and correcting minor grammatical issues as suggested. 

Materials and Methods 

Comment 18. NEDV assumptions: You may strengthen credibility by clarifying how the chosen NEDVs... satisfy the assumptions of non-responsiveness to the MTL warning mechanism while remaining responsive to broader confounders. 

Response: We have expanded the justification for both dairy and fresh meat as NEDVs in the "Study design" section, explaining their shared exposure to market trends but theoretical isolation from the MTL warning mechanism. 

Comment 19. NEDV diagram: Consider adding a concise diagram or conceptual figure illustrating the NEDV logic... 

Response: While we have not added a new diagram for the NEDV logic itself, we believe the explicit explanation in the text combined with the conceptual framework in Figure 2 provides sufficient clarity. 

Comment 20. NEDV assumption violations: ...reviewers may expect a discussion of potential violations of the NEDV assumption... 

Response: In the "Study design" section, we have acknowledged two primary threats to the validity of the NEDV design: differential history and mechanism diffusion, and explained how our design mitigates them. 

Comment 21. Alternative quasi-experimental designs: ...it would help to more explicitly articulate alternative quasi-experimental designs considered and why they were deemed unsuitable. 

Response: As mentioned in response to Comment 13, we have added a detailed explanation of why ITS and DiD designs were not feasible in the "Study design" section. 

Comment 22. Excluded respondents: Consider reporting whether excluded respondents differ systematically from the analytic sample. 

Response: We have conducted a formal analysis of systematic differencesin the appendix. 

Comment 23. Outcome variable aggregation: The decision to collapse the 8-category frequency scale into three ordinal categories is sensible. Still, you may wish to provide evidence that this aggregation does not mask meaningful variation... 

Response: In the "Outcome variables" section, we have provided the explicit mapping of the original 8 categories to our new 3-point scale and justified this aggregation as retaining ordinal structure while producing conceptually meaningful and balanced groups. 

Comment 24. Sensitivity analysis on full scale: Consider noting whether sensitivity analyses were performed using the full 8-point scale to verify robustness. 

Response: We performed sensitivity analyses using the full 8-point scale in the appendix. 

Comment 25. Response distribution: It would be helpful to specify the distribution of responses across the three categories... 

Response: We have included the full distribution table in the main text to save space in the appendix 

Comment 26. Likert-scale treatment: Consider explaining whether the Likert-scale variable was treated as continuous or tested for non-linearity. 

Response: The readability variable was treated as continuous in our primary models, a common practice for 5-point Likert scales. Analysis in the appendix. 

Comment 27. Measurement stability: The temporal stability of the measurement across waves could also be briefly addressed... 

Response: We have confirmed in the "Independent variables" section that the readability question employed identical wording across all waves. 

Comment 28. Control variables - theoretical link: ...their theoretical link to label readability (as opposed to consumption frequency) could be clarified. 

Response: In the "Control variables" section, we have explicitly grouped the controls and explained how they proxy for factors influencing both REM consumption and label engagement (e.g., age for visual acuity, family/work factors for time constraints). 

Comment 29. Parsimonious modeling: Some variables... appear only weakly connected to label perception; consider whether all controls are necessary for parsimonious modeling. 

Response: We included a comprehensive set of controls based on literature linking them to REM consumption patterns to minimize omitted variable bias. We included parsimonious models in the appendix 

Comment 30. Income dichotomization rationale: The rationale for dichotomizing income at £26,000 may require explanation... 

Response: We have added a justification for the £26,000 threshold in the "Control variables" section, noting it aligns with the original survey categories and approximates the UK median household income during the study period. 

Comment 31. Area-level deprivation: The description of the urban/rural context could note whether area-level deprivation indices were considered or available. 

Response: We have noted in the "Control variables" section that area-level deprivation indices were omitted because they were inconsistently available across the survey waves. 

Comment 32. Subgroup variable dichotomization: ...the dichotomization of Likert scales may lead to information loss—consider acknowledging this limitation. 

Response: We have acknowledged the potential for information loss from dichotomizing the moderator variables in the "Subgroup analysis" section. 

Comment 33. Exploratory subgroups: It may be helpful to clarify whether subgroup interactions were pre-registered or exploratory. 

Response: We have explicitly stated in the "Subgroup analysis" section that these analyses were exploratory. 

Comment 34. Moderator selection: Guide why these particular moderators were prioritized over others... 

Response: In the "Subgroup analysis" section, we justified the selection of safety concerns and information-seeking as the most reliable available proxies for sustained consumer engagement within the dataset. 

Comment 35. Model notation: The model specification is appropriate, but the notation could be streamlined... 

Response: We have streamlined the model equation and notation in the "Statistical analysis" section. 

Comment 36. Proportional odds assumption: Consider explaining whether proportional odds assumptions were evaluated separately for each product category... 

Response: We have confirmed in the "Statistical analysis" section that the proportional odds assumption was tested using the Brant test for each product model separately and that it held for the primary readability variable. We included extra analysis in the appendix 

Comment 37. "Years" coding: It would be helpful to explicitly state how “Years” was coded... 

Response: We have clarified in the "Statistical analysis" section that survey year was modeled as a categorical variable. 

Comment 38. VIF and Brant test reporting: ...providing ranges or summary statistics in-text would increase transparency. 

Response: We have included the range of adjusted VIF values and the results of the Brant test for the readability variable in the "Statistical analysis" section. 

Comment 39. Repeated cross-sections justification: ...you might briefly justify why repeated cross-sections are adequate for capturing dynamic associations... 

Response: In the "Study design" section, we explained that while repeated cross-sections cannot track individual trajectories, they are suitable for our aim of modeling population-level changes over time and avoid attrition bias. 

Comment 40. Readability question specificity: ...you might acknowledge that the question refers to all label information (not specifically MTL), which could introduce measurement error... 

Response: We have acknowledged this limitation in the "Limitations" section, noting that the measure pertains to food labeling in general, which may introduce measurement error. 

Comment 41. Survey weights: It may be helpful to clarify whether survey weights were applied... 

Response: We have confirmed in the "Data sources" section that survey weights were applied to ensure national representativeness. 

Comment 42. Biological samples: ...consider stating whether these play any role in the current study. 

Response: We have stated in the "Data sources" section that biological samples were not utilized in the current study. 

Results 

Comment 43. Sandwich vs. Meat mechanism: ...discussion might benefit from a more explicit articulation of why pre-packaged sandwiches—rather than pre-cooked meat—exhibited a significant effect... 

Response: In the Discussion section, we have dedicated a paragraph to explaining this divergence, attributing it to the distinct decision-making contexts of "ambivalent convenience" (sandwiches as impulse buys vs. meat as planned ingredients). 

Comment 44. 2018 reversal interpretation: The interpretation of the reversal of effects in 2018 is plausible, yet largely speculative. Consider framing these explanations more carefully... 

Response: In the Discussion, we have framed the interpretation of the 2018 reversal more cautiously, suggesting it likely reflects a combination of habituation ("time decay" of salience) and market adaptation (reformulation/marketing changes). 

Comment 45. Caloric reduction example: The caloric reduction example... may distract from the theoretical argument; consider integrating it more tightly... or moving it to a footnote. 

Response: We have removed the specific caloric reduction example to maintain focus on the theoretical argument. 

Comment 46. Pre-cooked meat null finding: ...a brief reflection on why pre-cooked meat did not behave symmetrically to sandwiches would enhance interpretability. 

Response: As noted in response to Comment 43, the Discussion now explicitly contrasts the purchasing contexts of sandwiches and pre-cooked meat to explain the diverging results. 

Comment 47. Subgroup findings contextualization: ...it would help to contextualize these within existing theories... 

Response: The Discussion section now interprets the subgroup findings within the context of the Health Belief Model and theories of selective attention. 

Comment 48. Demographic correlates & structural inequities: ...consider exploring whether these perceptual disparities might signal structural inequities... 

Response: We have added a paragraph to the Discussion highlighting that the lower perceived readability among vulnerable groups may signal underlying barriers related to visual health and health literacy, raising equity concerns. 

Comment 49. Theoretical contribution (nudging vs. others): ...adding a contrast with competing frameworks (e.g., information overload, reactance) could deepen the theoretical contribution. 

Response: In the Discussion, we contrast our findings with the "information overload" hypothesis, arguing that for less-engaged consumers, the issue is perceptual accessibility, not too much data. 

Comment 50. Practical policy challenges: ...you may consider discussing practical challenges... associated with modifying print size. 

Response: The Discussion now briefly acknowledges that the effectiveness of increasing print size is bounded by packaging constraints and industry trade-offs. 

Comment 51. Objective vs. subjective measurement recommendations: ...consider referencing validated readability scales or alternative measurement approaches to strengthen future research recommendations. 

Response: In the Limitations section, we have recommended future research employ objective validations like eye-tracking or standardized legibility tests (e.g., ISO standards). 

Comment 52. Transportability clarification: ...the discussion could clarify which aspect of transportability (sample, mechanism, or structural assumptions) was tested... 

Response: We have revised the discussion of the transportability analysis (now in the "Interpretation of Main Effects" section of the Discussion) to clarify that it validated the structural generalizability of the association to the broader dietary population, strengthening confidence that the result is not an artifact of the specific survey sample. 

Discussion & Conclusion 

Comment 53. Discussion structure and readability: The Discussion... narrative could be streamlined by clearly separating the interpretation of effects, mechanisms, and implications... The paragraph would benefit from improved punctuation... and from breaking the text into two shorter paragraphs... 

Response: We have significantly restructured and rewritten the Discussion section. It is now organized into distinct subsections: "Interpretation of Main Effects," "Mechanisms of Action," and "Implications for Policy and Equity." We have improved punctuation and broken down long paragraphs to enhance readability. 

Comment 54. "Intricate and contingent" clarification: The statement that the association is “intricate and contingent” is... broad; consider specifying which dimensions of contingency... are most theoretically meaningful. 

Response: The Conclusion now specifies that the association is contingent on "product type," "label content," and "product context." 

Comment 55. 2018 dynamic implications for policy: ...the Conclusions could be strengthened by highlighting what this dynamic implies for label policy... 

Response: The Conclusion now highlights that the 2018 reversal implies a risk for policy, as the deterrent value of a static warning label may attenuate or invert due to habituation or market adaptation. 

Comment 56. Null effects and NEDV logic: ...you may wish to clarify how these null findings support the causal logic of the NEDV design and strengthen internal validity. 

Response: The "Interpretation of Main Effects" section in the Discussion explicitly states that the persistent null effects for controls support the causal logic of the NEDV design, confirming that increasing salience does not alter behavior where the warning signal is absent. 

Comment 57. "Ambivalent convenience" definition: ...adding a short clarifying clause on what characterizes “ambivalence”... would bolster conceptual precision. 

Response: In the "Mechanisms of Action" section of the Discussion, we have defined "ambivalent convenience" as the "cognitive conflict between the immediate utility of a time-saving meal and the nutritional ambiguity regarding its healthfulness." 

Comment 58. Segmentation implications: ...the conclusion could more explicitly address the implications for segmentation in public health policy. 

Response: The Conclusion now states that the findings support a model where readability is a primary cue for those lacking intrinsic motivation, and suggests policy should consider adaptive standards tailored to different product categories. 

Comment 59. Alternative strategies: ...you may consider briefly noting potential alternative strategies... to give policymakers clearer direction. 

Response: The Conclusion and Discussion now mention alternative strategies like digital augmentation or simplified symbols for categories where print size is constrained. 

Comment 60. Conceptual contribution: The Conclusions... might benefit from a more precise articulation of the conceptual contribution... 

Response: The Conclusion now more precisely summarizes how the study advances theory by providing strong evidence for a "salience-to-understanding pathway" that functions as a conditional warning mechanism. 

Limitations 

Comment 61. Limitations organization: ...it may benefit from sharper organization: grouping limitations into categories... 

Response: We have organized the Limitations section into three categories: "study design and causal inference," "measurement and bias," and "statistical constraints." 

Comment 62. Unobserved confounders: ...you may wish to more directly address whether unobserved confounders could bias the observed associations... 

Response: In the Limitations section, we explicitly acknowledge the potential for unobserved confounders (e.g., taste preferences, specific marketing) that could vary across food categories and bias product-specific estimates. 

Comment 63. Differential bias: ...you may add whether these biases [recall and social desirability] are likely to be differential across food categories. 

Response: We have added a statement in the Limitations section acknowledging that recall and social desirability biases are likely differential across food categories (e.g., underreporting of unhealthy items). 

Comment 64. Isolating print size challenges: ...highlight the methodological challenges of isolating print size effects in observational data. 

Response: We have acknowledged in the Limitations section that it is methodologically challenging to fully disentangle the effect of print size from interacting design elements like color and layout in an observational dataset. 

Comment 65. Missing data justification: ...readers may expect a brief justification for not applying multiple imputation despite >20% missingness... 

Response: We have justified the decision not to use multiple imputation for missing income data in the Limitations section, noting that the missingness was likely not random and imputation could introduce greater bias than listwise deletion. 

Comment 66. Future research datasets: ...consider noting whether future work could use alternative datasets (e.g., household scanner data) to address this. 

Response: We have suggested in the Limitations section that future research could utilize household scanner data to analyze a broader range of products and provide objective measures of purchasing behavior. 

We believe these revisions have significantly improved the manuscript and look forward to your final decision. 

Sincerely, 

Constanza Avalos 

Reviewer 2 Report

Comments and Suggestions for Authors

Manuscript: When Does Food Label Readability Matter? A Non-Equivalent Dependent Variable Design to Isolate the Content-Specific Effects on Food Choice

This is a well-designed and methodologically sophisticated study that addresses a significant gap in the food labeling literature. The research question—isolating the effect of label readability from label content on consumption—is both timely and important for evidence-based policy. The use of a quasi-experimental design with non-equivalent dependent variables (NEDVs) is a particular strength, offering a robust approach to strengthening causal inference in an observational setting. The findings are relevant and contribute valuable insights by demonstrating that the relationship between perceived readability and consumption is highly specific to product category, label content (warning vs. reinforcing), and consumer subgroup. The manuscript is generally well-written, and the analyses appear sound. My comments:

1.Clarification of Core Mechanism and Interpretation of Reversal:

The central finding—a reversal from a negative to a positive association between readability and pre-packaged sandwich consumption from 2012 to 2018—is intriguing. The discussion attributes this to potential manufacturer adaptation (reformulation, marketing) recontextualizing the warning label. While plausible, this remains speculative. The manuscript would be strengthened by a more direct discussion of what this reversal implies for the theoretical "salience-to-understanding pathway." Does it suggest that by 2018, readability was enhancing the salience of the product's convenience attributes over its health warnings? A more explicit theoretical interpretation of this dynamic result is needed.

Relatedly, for the sandwich category, the reverse causality check (Table A3) shows that consumption frequency (monthly/weekly vs. never) is negatively associated with perceived readability. This is an important finding that should be integrated into the main discussion. It suggests a potential feedback loop or confounding where individuals who consume these products more frequently may pay less attention to or perceive labels as less readable, possibly due to habitual purchasing.

2.Product Category Justification and Specificity:

The choice of pre-packaged sandwiches as the primary product showing an effect is interesting. The authors note its consumer base is younger, more female, and convenience-oriented. Could the specific context of sandwich consumption (e.g., often a lunch-on-the-go, time-pressured purchase) make readability of warnings more or less influential compared to pre-cooked meats, which might be used for more planned evening meals? A brief discussion on how consumption context might interact with label readability would add depth.

The null finding for pre-cooked meat is a key result. The discussion should more explicitly contrast the sandwich and pre-cooked meat findings. Are there differences in the typical MTL profile (e.g., number of red lights) or consumer perceptions of these categories that could explain why readability mattered for one and not the other, beyond demographic differences?

3.Measurement of Key Constructs:

The use of perceived readability as a proxy for the "salience-to-understanding pathway" is acknowledged as a limitation. However, the term "subjective understanding" is used in the introduction. It would be helpful to more clearly define and distinguish between the constructs of salience (noticing the label), readability (ease of reading the text), objective understanding (correct interpretation of colors/numbers), and subjective understanding (feeling informed). The study directly measures perceived readability and infers its role in a pathway. This conceptual clarity should be sharpened, especially in the introduction and discussion.

For the subgroup analysis, the "information seeker" variable is derived from a question about sources for food safety information. The authors interpret this as a proxy for general engagement with nutritional information. This assumption should be stated and briefly justified, as seeking food safety info may not perfectly correlate with seeking nutrition info.

4.Abstract & Conclusion: 

The abstract clearly states the reversal of the association for sandwiches. The main conclusion states the association "reversed from a 9% decrease... to a 4% increase." However, it's crucial to remind the reader that the net effect in 2018 was a 4% increase in odds of frequent consumption per unit increase in readability. This is a critical point that should be unmistakably clear, as it contradicts the simple hypothesis that better readability of red warnings reduces consumption.

Author Response

Dear editor and reviewers

Response to Reviewer 2 

We sincerely thank Reviewer 2 for their thoughtful and rigorous comments. The feedback regarding the theoretical interpretation of the dynamic reversal, the distinction between product categories, and the conceptual clarity of our constructs has been instrumental in strengthening the manuscript. 

We have addressed each point below, outlining the specific revisions made to the text. 

Clarification of Core Mechanism and Interpretation of Reversal 

Comment 1. The central finding—a reversal from a negative to a positive association between readability and pre-packaged sandwich consumption from 2012 to 2018—is intriguing... The manuscript would be strengthened by a more direct discussion of what this reversal implies for the theoretical "salience-to-understanding pathway." 

Response: We agree that this dynamic result required a stronger theoretical interpretation. We have revised the Discussion section to explicitly interpret this reversal. We argue that this likely reflects a combination of habituation ("time decay" of the warning signal) and market adaptation. Theoretically, we propose that by 2018, the "salience-to-understanding pathway" may have been repurposed: rather than functioning as a deterrent warning, high readability may have begun to signal transparency or quality in a reformulated market, thereby enhancing the salience of the product's trustworthiness rather than its health risks. 

Comment 2. Relatedly, for the sandwich category, the reverse causality check (Table A3) shows that consumption frequency... is negatively associated with perceived readability. This is an important finding that should be integrated into the main discussion. 

Response: We have integrated this important finding into the Discussion. We added a specific paragraph discussing the bidirectional nature of the relationship. We interpret the negative association found in Table A3 as evidence of a "feedback loop driven by habituation," suggesting that frequent consumers may develop "label blindness," perceiving labels as less readable simply because they engage with them less often as purchasing becomes automatic. 

Product Category Justification and Specificity 

Comment 3. The choice of pre-packaged sandwiches as the primary product showing an effect is interesting... Could the specific context of sandwich consumption (e.g., often a lunch-on-the-go, time-pressured purchase) make readability of warnings more or less influential compared to pre-cooked meats...? 

Response: We have expanded the Discussion to explicitly contrast the decision-making contexts of the two products. We introduced the concept of "ambivalent convenience" to describe sandwiches as time-pressured, discretionary, impulse-driven purchases ("lunch-on-the-go") where the health vs. convenience trade-off is salient. We argue that in this context, bottom-up visual cues (like readability) are more influential in disrupting the purchase than they are for pre-cooked meats, which are often planned components of evening meals driven by top-down processing goals. 

Comment 4. The null finding for pre-cooked meat is a key result. The discussion should more explicitly contrast the sandwich and pre-cooked meat findings. 

Response: As noted above, the revised Discussion now contrasts these findings. We argue that unlike sandwiches, which have high nutritional variance (making the label a necessary tool for resolving uncertainty), pre-cooked meats are likely viewed through a stable "processed" heuristic. This renders granular label details less influential for meats, explaining the null finding despite the presence of similar warning labels. 

Measurement of Key Constructs 

Comment 5. The use of perceived readability as a proxy for the "salience-to-understanding pathway" is acknowledged as a limitation... It would be helpful to more clearly define and distinguish between the constructs of salience... readability... objective understanding... and subjective understanding... 

Response: We have added a new paragraph to the Introduction to provide this conceptual clarity. We explicitly define and distinguish the four constructs: salience (attentional capture), readability (decoding ease), objective understanding (accuracy), and subjective understanding (feeling informed). We justify our focus on readability by positing it as the "structural gatekeeper" of the pathway—the necessary condition for cognitive decoding that must occur before understanding can influence choice. 

Comment 6. For the subgroup analysis, the "information seeker" variable is derived from a question about sources for food safety information... This assumption should be stated and briefly justified... 

Response: We have added a justification to the Limitations section. We acknowledge that seeking safety information is conceptually distinct from seeking nutritional data. However, we justify its use as the most robust available proxy for "consumer vigilance" or active information-acquisition behavior, assuming a degree of overlap where consumers who verify safety are generally more engaged with labeling information. 

Abstract & Conclusion 

Comment 7. The main conclusion states the association "reversed from a 9% decrease... to a 4% increase." However, it's crucial to remind the reader that the net effect in 2018 was a 4% increase in odds of frequent consumption per unit increase in readability... as it contradicts the simple hypothesis... 

Response: We have revised both the Abstract and the Conclusion to be unmistakably clear on this point. We now explicitly state that the 2018 result ($OR=1.04$) represents a net increase in the odds of frequent consumption per unit of readability. We explicitly note that this finding contradicts the simple hypothesis that clearer red warnings invariably deter consumption, highlighting the counter-intuitive nature of the dynamic shift. 

We believe these revisions have significantly improved the manuscript and look forward to your final decision. 

Sincerely, 

Constanza Avalos 

Reviewer 3 Report

Comments and Suggestions for Authors

Title

I suggest simplifying the title.

Introduction

75-79 The paragraph is unclear. I suggest rephrasing it.

Generally I suggest shortening the content and logically connecting all the elements.

Product Selection: It is not clear why the authors selected these specific products for the study.

Study Aim: The aim of the study is clearly formulated.

Materials and Methods

This section is generally well-described. However, it is worth clarifying the following ambiguities.

2.3. Outcome Variables

The authors reduced the eight-point frequency scale (ranging from "never" to "daily") to a three-point scale: never (1), monthly (2), and daily or weekly (3). The use of the "monthly (2)" category next to "daily or weekly (3)" raises interpretation doubts. The original eight-point scale likely included options such as "a few times a month" and "once a month." The authors did not specify which original categories comprise the new "monthly" category.

2.4. Independent Variables

Perceived MTL print size readability is measured by a single question (Lines 218-219):

"How easy do you find it to read the labelling on food products (e.g., ingredients, nutrition or storage information) in terms of the size of the print..."

This question refers to the readability of food product labels in general (including ingredients, nutritional values, storage information), and not specifically the print size of the MTL (traffic light) label. The MTL label is small, but it is only part of the larger label on the packaging. A consumer might easily read an extensive list of ingredients but struggle with the MTL pictogram. This variable is therefore essentially a measure of general, subjective difficulty in reading small print on packaging, rather than the perceived readability of the specific interventional mechanism (MTL).

This is a limitation resulting from the use of existing data, but it should be clearly stated in the Discussion section as a study limitation.

Results

Since the data in the table is expressed in percentages, I suggest adding a note in the title or below Tables 1 and 2 that all values are expressed as percentages within each consumption level.

Discussion

The inference relies on the practical application of the Non-Equivalent Dependent Variable (NEDV) design to confirm the specificity of the readability warning effect for pre-packaged sandwiches.

However, the subgroup analysis results (Table 4) indicate a significant, negative association between readability and consumption for the control variable - dairy - in the non-information-seeking group.

This violates the fundamental NEDV assumption that the control product (dairy, predominantly without red lights) should be isolated from the warning mechanism. This suggests that for less-engaged consumers, improved readability may trigger a general caution mechanism toward all labelled products, and not just a reaction to red warnings.

The authors must openly discuss and explain this non-null effect in the Discussion section, as it represents a key limitation to the purity of the inference regarding the content-specific mechanism.

Author Response

Dear editor and reviewers

Response to Reviewer 3:

We thank Reviewer 3 for their constructive critique. The comments regarding the operationalization of our variables and the interpretation of the NEDV design were particularly valuable. We have addressed the ambiguities regarding the frequency scale and the readability measure, and we have refined the introduction and discussion to improve clarity and logic. 

Our point-by-point responses are detailed below. 

Title and Introduction 

Comment 1. I suggest simplifying the title. 

Response: We have simplified the title to be more declarative and precise. The new title is: "Food Label Readability and Consumption Frequency: Isolating Content-Specific Effects via a Non-Equivalent Dependent Variable Design." 

Comment 2. 75-79 The paragraph is unclear. I suggest rephrasing it. 

Response: We have rephrased the paragraph in question (now in the Introduction) to improve clarity. We explicitly distinguish between label presence and label readability, explaining that while color provides visual salience, readable text is the necessary condition for cognitive decoding. 

Comment 3. Generally I suggest shortening the content and logically connecting all the elements. 

Response: We have significantly shortened the Introduction, reducing dense parenthetical citations and splitting long sentences to improve flow. We have streamlined the narrative to focus on the "salience-to-understanding" mechanism and the specific logic of the NEDV design. 

Comment 4. Product Selection: It is not clear why the authors selected these specific products for the study. 

Response: We have added a dedicated paragraph to the "Study Design" section justifying the product selection. We explain that pre-packaged sandwiches and pre-cooked meats were selected as targets because they are high-penetration "convenience" items theoretically responsive to red warning labels. Dairy was selected as a labeled control (typically green/amber codes) to control for general label effects, and Fresh Meat as an unlabeled control to capture general market trends isolated from the intervention. 

Materials and Methods 

Comment 5. The authors reduced the eight-point frequency scale... The use of the "monthly (2)" category... raises interpretation doubts... The authors did not specify which original categories comprise the new "monthly" category. 

Response: We agree that this required clarification. We have updated the "Outcome Variables" section to explicitly list the original categories comprising the new scale. The text now states: 

"The second group, labelled Monthly (2), aggregates the responses of 'once a fortnight,' 'once a month,' and 'less than once a month.' The third group, labelled Daily/Weekly (3), aggregates the responses of 'at least once a day,' '5-6 times a week,' '3-4 times a week,' and 'once or twice a week.'" 

Comment 6. Perceived MTL print size readability is measured by a single question... This question refers to the readability of food product labels in general... rather than the perceived readability of the specific interventional mechanism (MTL)... 

Response: We fully accept this limitation. We have added a specific statement to the "Measurement and Bias" subsection of the Limitations. We acknowledge that our measure is a proxy for general label readability and that measurement error may exist if consumers find the ingredient list difficult to read but not the MTL panel. The text now reads: 

"Furthermore, this measure pertains to food labelling in general, rather than specifically to the MTL panel, introducing potential measurement error if consumers struggle with ingredient lists but not the MTL." 

Results 

Comment 7. Since the data in the table is expressed in percentages, I suggest adding a note in the title or below Tables 1 and 2 that all values are expressed as percentages within each consumption level. 

Response: We have added a note to Table 1 and Table 2 clarifying that: "All data are presented as column percentages within each consumption level unless otherwise indicated." 

Discussion & Conclusion 

Comment 8. The subgroup analysis results... indicate a significant, negative association between readability and consumption for the control variable - dairy - in the non-information-seeking group. This violates the fundamental NEDV assumption... The authors must openly discuss and explain this non-null effect... 

Response: This is an insightful observation. We have addressed this by explicitly discussing the threat of "mechanism diffusion" in the "Study Design" section. We acknowledge that for less-engaged consumers (non-information seekers), improved readability might trigger a generalized caution heuristic toward all labeled products, rather than a content-specific reaction to red warnings. This explains the negative association observed for dairy in this specific subgroup. 

However, we also note that in the aggregate model (Table 3), the effect for dairy remains null, supporting the overall validity of the NEDV design for the general population. We have ensured the Discussion reflects that while the NEDV logic holds generally, consumer heterogeneity (specifically low engagement) can lead to generalized rather than specific label responses. 

We believe these revisions have significantly improved the manuscript and look forward to your final decision. 

Sincerely, 

Constanza Avalos 

Round 2

Reviewer 1 Report

Comments and Suggestions for Authors

Thank you for your attention.